# The Cross-Regulation Between Set1, Clr4, and Lsd1/2 in *Schizosaccharomyces pombe*

**Haoran Liu[1], Bahjat Fadi Marayati[2], David de la Cerda[1], Brendan Matthew Lemezis[1], Jieyu Gao[1], Qianqian Song[3], Minghan Chen[4], Ke Zhang Reid[1]***

**1** Department of Biology, Wake Forest University, Winston-Salem, North Carolina, United States of America,
**2** Department of Biochemistry, Duke University School of Medicine, Durham, North Carolina, United States of America, **3** Department of Health Outcomes and Biomedical Informatics, University of Florida, Gainesville, Florida, United States of America, **4** Department of Computer Science, Wake Forest University, Winston-Salem, North Carolina, United States of America

* zhangk@wfu.edu

**Data Availability Statement:** All relevant data are within the manuscript and its Supporting information files. Genome-wide data, including RNA-seq, and ChIP-seq data are available in the

## Abstract

Eukaryotic chromatin is organized into either silenced heterochromatin or relaxed euchromatin regions, which controls the accessibility of transcriptional machinery and thus regulates gene expression. In fission yeast, *Schizosaccharomyces pombe*, Set1 is the sole H3K4 methyltransferase and is mainly enriched at the promoters of actively transcribed genes. In contrast, Clr4 methyltransferase initiates H3K9 methylation, which has long been regarded as a hallmark of heterochromatic silencing. Lsd1 and Lsd2 are two highly conserved H3K4 and H3K9 demethylases. As these histone-modifying enzymes perform critical roles in maintaining histone methylation patterns and, consequently, gene expression profiles, cross-regulations among these enzymes are part of the complex regulatory networks. Thus, elucidating the mechanisms that govern their signaling and mutual regulations remains crucial. Here, we demonstrated that C-terminal truncation mutants, *lsd1-ΔHMG* and *lsd2-ΔC*, do not compromise the integrity of the Lsd1/2 complex but impair their chromatin-binding capacity at the promoter region of target genomic loci. We identified protein-protein interactions between Lsd1/2 and Raf2 or Swd2, which are the subunits of the Clr4 complex (CLRC) and Set1-associated complex (COMPASS), respectively. We showed that Clr4 and Set1 modulate the protein levels of Lsd1 and Lsd2 in opposite ways through the ubiquitin-proteasome-dependent pathway. During heat stress, the protein levels of Lsd1 and Lsd2 are upregulated in a Set1-dependent manner. The increase in protein levels is crucial for differential gene expression under stress conditions. Together, our results support a cross-regulatory model by which Set1 and Clr4 methyltransferases control the protein levels of Lsd1/2 demethylases to shape the dynamic chromatin landscape.

## Author summary

Histone-modifying enzymes make covalent modifications to histones. These modifications act like chemical tags that can either tighten or loosen the DNA structure, affecting whether genes are turned on or off. In fission yeast, Set1 methylates histone H3 lysine 4,

Gene Expression Omnibus with accession number GSE246595 and GSE246596.

**Funding:** KZR received a federal grant from the National Institute of General Medical Sciences (R15GM 139107-01). The funder's website: https://www.nigms.nih.gov/. Q.S. is supported by the National Institute of General Medical Sciences (R35GM151089). The funder's website: https://www.nigms.nih.gov/. JG was supported by the URECA Summer Fellowship from Wake Forest University (https://ureca.wfu.edu/). BFM and BML obtained Fellowships from the Center for Molecular Signaling at Wake Forest University (https://molecularsignaling.wfu.edu/). The funders had no role in the study design, data collection, and analysis, decision to publish, or preparation of the manuscript.

**Competing interests:** The authors have declared that no competing interests exist.

which marks loosely packed DNA and is associated with gene activation; while Clr4 methylates histone H3 lysine 9, which represents a hallmark of the tightly packed DNA and is associated with gene silencing. Here, we show a regulatory relationship between these two enzymes and with two lysine-specific demethylases (Lsd1/2), which can remove the methyl tags added by Clr4 and Set1. Clr4 and Set1 have opposite effects on Lsd1 and Lsd2 protein levels. Clr4 reduces the levels of Lsd1/2, while Set1 promotes their stability. This control over the levels of Lsd1/2 is achieved through the ubiquitin-proteasome-dependent pathway. By studying these interactions, we have uncovered a novel regulatory mechanism that helps fission yeast maintain a balanced level of these active/repressive histone-modifying enzymes. Understanding these intricate regulatory networks offers important insights into gene control and enhances our comprehension of the complex interactions within cells.

## Introduction

The compartmentalization of eukaryotic genomes inside the nucleus presents a complex packaging paradigm whereby DNA is wrapped around histones, forming nucleosomes, and further packaged into higher-order chromatin [1–6]. Chromatin is organized into either closed (silenced) heterochromatin or open (active) euchromatin domains, which control the accessibility of the transcriptional machinery and, hence, regulate gene expression [7,8]. Histones, including the H3, H4, H2A, and H2B subunits, form the core particle of the nucleosome [1,9–11]. The N-terminal tails of histones are post-translationally modified by various covalent modifications such as methylation, acetylation, and phosphorylation [12–14]. Histone-modifying enzymes that add or remove these modifications regulate the dynamic landscape of chromatin [14–20]. One of the most well-studied post-translational histone modifications is the methylation of multiple lysine (K) residues on histones H3 and H4 [21,22]. Even though the methylation of different histone lysine residues may appear chemically similar, the functional consequences of these methylations are complex and associated with distinct chromatin organization [23]. For example, methylation of histone H3 lysine 9 (H3K9me) is globally associated with heterochromatin and gene silencing, while the methylation of histone H3 lysine 4 (H3K4me) is an indication of euchromatin and active gene expression [24,25]. Therefore, the proper regulation and turnover of histone-modifying enzymes play essential roles in gene expression, genome stability, and cell fate determination [26–28]. The cross-regulation between these enzymes ensures that their activities are fine-tuned and produces a flexible regulatory circuit that achieves cellular homeostasis. Dysregulation of histone modifiers can lead to profound deviations from normal physiological conditions. Specifically, the mis-regulation of these modifiers has been implicated in the pathogenesis of chronic neurological disorders and cancer [23].

Controlled protein degradation is vital for the maintenance of protein homeostasis [29,30]. The ubiquitin-proteasome system (UPS) is the major proteolytic system in eukaryotes that marks proteins with a 76-amino acid molecule: ubiquitin [31]. This degradation pathway is mediated by a multi-step cascade reaction encompassing E1, E2, and E3 enzymes [32–35]. Specifically, the E1 enzymes activate ubiquitin utilizing ATP hydrolysis and transfer it to E2 [36–38]. The E2 enzymes continue to pass the activated ubiquitin to the substrates through interactions with the E3 ligases [37–41]. Compared with E1 and E2, E3 ligases take on more responsibility to recognize the substrates and ensure the ubiquitination of the correct proteins [42,43]. The target proteins are shuttled to the proteasome, which catalyzes the cleavage of the proteins

to amino acids and small peptides that can be recycled for new protein synthesis [39,44,45]. Previous studies have observed that about 43% of the methylation sites of proteins in budding yeast, *Saccharomyces cerevisiae* (*S. cerevisiae*), could be potential target sites for ubiquitination [46], which indicates the possibility of protein methylation competing with ubiquitination to protect proteins from degradation.

In fission yeast, *Schizosaccharomyces pombe* (*S. pombe*), Set1 is the sole catalytic unit of macromolecular complex called COMPASS (Complex Associated with Set1) [47,48]. Set1 is responsible for mono-, di- and tri-methylation on H3K4 and is often recruited to the 5′ end near the transcriptional start site (TSS) of actively transcribed genes and associates with transcription elongation machinery [49–52]. Notably, the H3K4 methylation is stimulated by the upstream ubiquitination of H2B K119, which is achieved by HULC (Histone H2B Ubiquitin Ligase Complex) [53,54]. The ubiquitination of H2B leads to a conformational change in COMPASS, thereby promoting the catalytic activity of Set1 by inhibiting the blockage of catalytic modules [55–59].

In *S. pombe*, the sole H3K9 methyltransferase, Clr4, initiates H3K9 methylation to silence heterochromatin [60,61]. Clr4 forms a protein complex named CLRC (Clr4 methyltransferase Complex), and all components are required for heterochromatin silencing [62,63]. The SET (Su(var)3-9, Enhancer-of-zeste and Trithorax) domain of Clr4 allows it to mediate the "writing" mechanism for heterochromatin assembly at pericentromeric repeats, sub-telomeric regions, and the mating type locus [60,63]. The chromodomain of Clr4 recognizes and binds to methylated H3K9 and promotes the maintenance and spreading of heterochromatin across large chromatin domains [63]. Heterochromatin is usually correlated with transcriptional or post-transcriptional gene silencing, which limits the accessibility of RNA polymerase and restricts the accumulation of heterochromatic transcripts, respectively [64,65]. Loss of Clr4 leads to the loss of heterochromatin and causes severe gene silencing defects [66].

Histone methylation marks are reversible due to the existence of histone demethylases. LSD1/KDM1a was the first identified member of the FAD (flavin adenine dinucleotide)-dependent histone demethylase family in the mammalian system [67,68]. It is a highly conserved mono- and di-methylated H3K4/H3K9 demethylase, which can function as a transcriptional repressor when it removes the methyl groups from H3K4 or a gene activator when it removes the methyl groups from H3K9 in association with the androgen receptor [69]. LSD2/KDM1b, the mammalian paralog of LSD1/KDM1a, mainly targets H3K4 for demethylation [70,71]. In fission yeast, Lsd1 forms a complex with its paralog Lsd2, and two plant homeodomain (PHD) finger proteins, Phf1 and Phf2 [72–74]. This complex, known as Lsd1/2, SWIRM1/2, or SAPHIRE complex, localizes specifically at heterochromatic loci and the transcription start sites of actively transcribed genes, which indicates its dynamic role in regulating gene expression [75]. Lsd1/2 complex also maintains the boundary between euchromatin and heterochromatin at the telomeres, presumably by differentially removing methylation marks from histone H3K9 and/or H3K4 [76]. The revelation of the Lsd1/2 complex prompts us to delve into the noncatalytic functions of both Lsd1 and Lsd2 and the roles of the proteins they are linked with. Nevertheless, the precise architecture of the Lsd1/2 complex remains to be determined.

The protein abundance and activity of Lsd1 and Lsd2 are governed by a large and complicated regulation network [77]. However, the exact mechanism by which Lsd1/2 undergoes protein degradation remains unclear. Cul4, containing the highly conserved Cullin domain, plays an essential role in assembling the muti-subunit Cullin-RING E3 ubiquitin ligase (CRL) complexes [78,79]. In human cells, CUL4A and CUL4B interact with an adaptor protein DDB1 (DNA damage binding protein 1) and their associated factor DDB2 to promote ubiquitination on histones [80–82]. In *S. pombe*, Cdt2 is the Ddb1-and Cul4-associated factor, and the function

of the Cul4-Ddb1$^{Cdt2}$ complex has also been reported in protein degradation [83,84]. For example, the degradation of ribonucleotide reductase inhibitor protein, Spd1, requires the Cul4-Ddb1$^{Cdt2}$ complex, which is crucial for genome stability and cell differentiation into meiosis [85,86]. Additionally, Cul4-Ddb1$^{Cdt2}$ directly recognizes Epe1, an anti-silencing factor and a potential H3K9 demethylase, to promote its polyubiquitination and degradation [87]. Cul4 is also part of CLRC and functionally correlates with two other components in CLRC: the WD-40 protein, Raf1; and the β-propeller protein, Rik1 [41,88]. Cul4-Rik1-Raf1 structurally resembles Cul4-Ddb1$^{Cdt2}$ E3 ubiquitin ligase and exhibits ubiquitin ligase activity *in vitro* [89,90].

In this study, we generated viable Lsd1/2 mutants with non-functional high-mobility-group (HMG) box domains at the C-terminus, allowing for convenient manipulation in the laboratory. We identified that the chromatin binding of the Lsd1/2 complex is dependent upon the HMG domain of Lsd1/2. Utilizing a yeast two-hybrid approach, we directly detected physical interactions among Lsd1, Phf1, and Phf2. While Lsd2 was co-purified with Lsd1, Phf1, and Phf2, we did not identify direct physical interactions between Lsd2 and Lsd1 or Phf2, suggesting a divergent/non-redundant role for Lsd2. We also uncovered interactions between Lsd1/2 and Raf2 or Swd2, subunits of CLRC or COMPASS complexes, respectively. Interestingly, we observed that Set1 and Clr4 have opposite effects on Lsd1 and Lsd2 protein levels, which is mediated through the ubiquitin-proteasome-dependent pathway. Without Set1, the protein levels of Lsd1/2 are reduced, while the absence of Clr4 enhances Lsd1/2 protein levels. Under heat stress, Set1 is essential for the upregulation of Lsd1/2; these increased protein levels are critical for the differential gene expression observed during heat stress. Additionally, our findings demonstrated that CLRC exerts an antagonistic effect by controlling the protein level of Set1. Set1 protein abundance is controlled by H2B ubiquitination, which facilitates the chromatin-associated activities of Set1 during heat stress. Overall, our results unfold a cross-regulatory mechanism in which the methyltransferases Set1 and Clr4 fine-tune the protein levels of the demethylases Lsd1/2, thus shaping the dynamic landscape of chromatin and maintaining cellular homeostasis.

## Results

### The C-terminal domains are essential for Lsd1 and Lsd2 chromatin binding

The complete loss of Lsd1 or Lsd2 results in severe growth defects or causes lethality in the cell, respectively [72,73,91]. Meanwhile, catalytic mutants in the amine oxidase domain of Lsd1 (from amino acids 267–775, the mutation is *K603AK604A*) and Lsd2 (from amino acids 516–1035, the mutation is *K823AK824A*) both show no apparent growth defects [91]. These results led us to wonder whether Lsd1/2 proteins have important functions beyond their amine oxidase catalytic activities. To investigate the non-catalytic functions of Lsd1 and Lsd2, we generated two C-terminal truncations named *lsd1-ΔHMG* (deletion of amino acids 841–1000 including the HMG domain, a High Mobility Group domain implicated in DNA binding) [92] and *lsd2-ΔC* (deletion of amino acids 1165–1235) (Fig 1A). Notably, deletion of the Lsd2 HMG-domain results in lethality akin to that in the complete loss of Lsd2, which further implies the unknown and important functions of the C-terminus of these two proteins. *lsd2-ΔC* has a shorter deletion of the C-terminus compared to the truncation of the Lsd2 HMG-domain, which may partially impair the HMG-domain of Lsd2. Both *lsd1-ΔHMG* and *lsd2-ΔC* show moderate growth and silencing defects and retain partial or full enzymatic activities (S1 Fig) [91], which demonstrates that the C-terminus, including the HMG domain, possesses functions that are independent of the amine oxidase-related catalytic activity. We tagged FTP (Flag-TEV-Protein A) at the C-terminus in wild-type or mutated Lsd1 and Lsd2 for further precipitation (IgG Sepharose beads) and detected their protein levels using peroxidase anti-

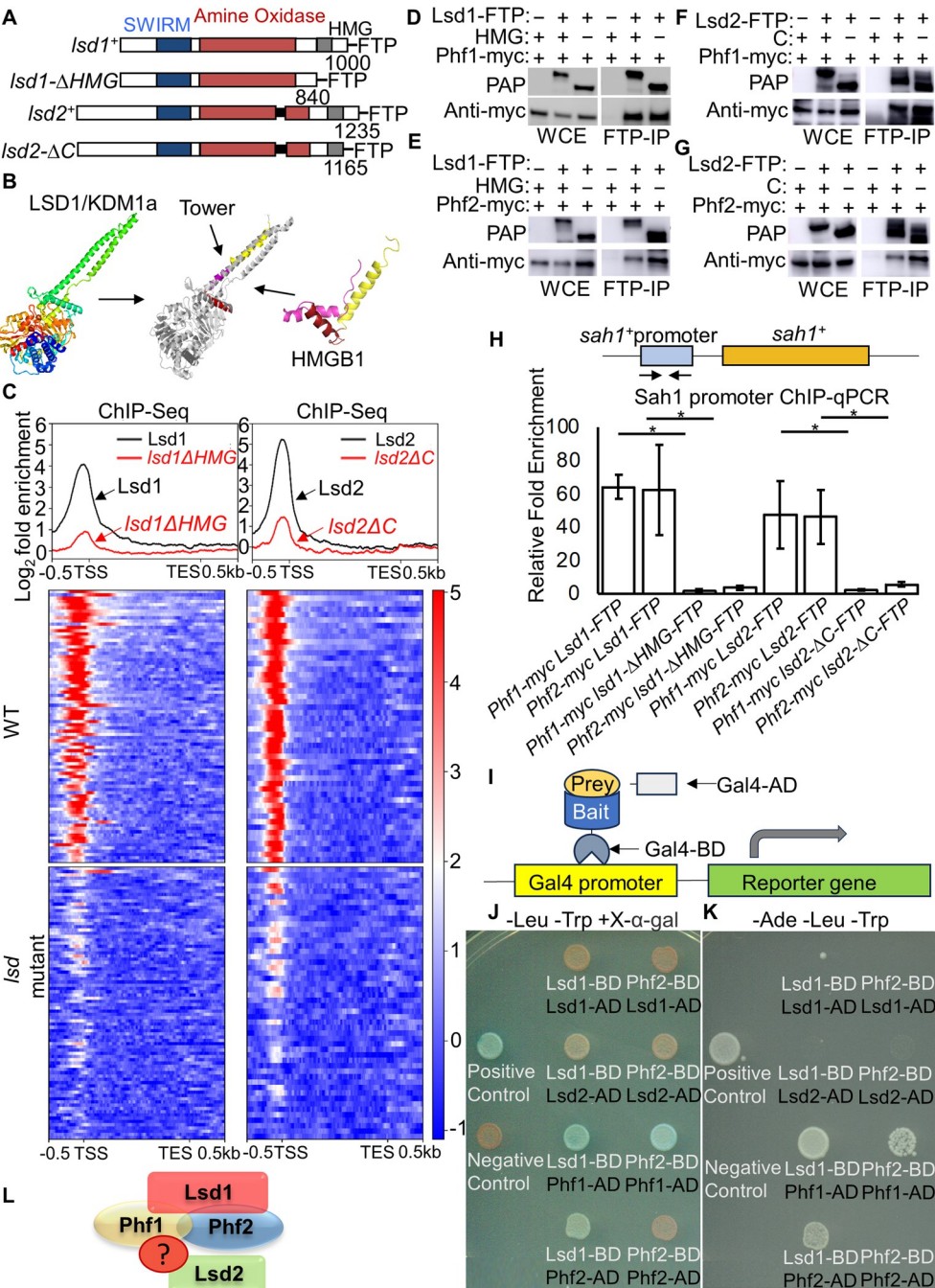

**Fig 1. The function of the C-terminal domains of Lsd1 and Lsd2 and the architecture of Lsd1/2 complex.** (A) Gene schematics for full-length Lsd1 and Lsd2 with previously described C-terminus truncation mutants (*lsd1-ΔHMG* and *lsd2-ΔC*). Protein domains of Lsd1 and Lsd2 are color-coded, and numerical labels represent the respective protein lengths. Introns in Lsd2 are marked in black. Lsd1, Lsd2, and their mutant forms are all tagged with FTP (Flag-TEV-Protein A) at the C-terminus. (B) Computational analysis of the crystal structure of mammalian LSD1/KDM1a (left, PDB ID: 2Z3Y) and HMG-box protein (HMGB1) (right, PDB ID: 1NHM). Overlapping crystal structures (middle) were generated using Phyre 2.0. (C) ChIP-Seq peak enrichments of Lsd1/2-FTP and their mutants are aligned with genomic regions spanning 500 base pairs upstream of the transcriptional starting site (TSS) to 500 base pairs downstream of the transcriptional termination site (TES). Heatmaps represent the binding of Lsd1/2-FTP and their mutants to the most robustly bound genes. Each line in the heatmap represents a gene. The color scale, from blue to red, indicates increasing enrichment of Lsd1/2-FTP and their mutants at specific genome loci compared to the untagged negative control. On the top, aligned Lsd1 or Lsd2 occupancy values for all selected genes are plotted as moving averages along their relative genomic positions. (D-G) Co-immunoprecipitation followed by western blotting

reveals protein-protein interactions between Lsd1 and Phf1 (D) as well as Phf2 (E). Similar interactions were observed between Lsd2 and Phf1 (F) and Phf2 (G). +/- signs indicate the presence or absence of the protein domain. IgG Sepharose beads were used for protein immunoprecipitation. WCE: whole cell extract. Peroxidase anti-peroxidase (PAP) was used in western blot to detect FTP signals. (H) ChIP-qPCR demonstrates the chromatin binding ability of Phf1/2 with or without the C-terminal domains of Lsd1/2 proteins. Phf1/2-myc was immunoprecipitated using an anti-myc antibody after preclearing Protein A with TEV cleavage. Asterisks indicate $p$-values $\leq 0.05$ as determined by a Student's $t$-test, comparing the indicated samples with WT values. Error bars represent the standard error of the mean (s.e.m.). Horizontal lines indicate significance between wild-type and mutants. (I) Cartoon illustrates the yeast two-hybrid approach. Gal4-BD is fused with the bait protein, and Gal4-AD is fused with the prey protein. When the bait protein interacts with the prey protein, Gal4-BD and AD come into close proximity, initiating reporter gene expression. (J-K) Verification of reporter gene expression and protein interactions between Lsd1, Phf1, and Phf2 using the yeast two-hybrid system. (J) Medium lacking leucine and tryptophan but containing X-α-gal (-Leu -Trp +X-α-gal). (K) Medium without adenine, leucine, and tryptophan (-Ade -Leu -Trp). (L) The cartoon demonstrates the direct interaction between Lsd1, Phf1, and Phf2 while indicating that Lsd2 does not directly interact with Lsd1 and Phf2.

peroxidase (PAP) (Fig 1A). The FTP-tagged wild-type Lsd1 and Lsd2 are fully functional, and cells carrying the wild-type FTP-tagged Lsd1 and Lsd2 have no growth defect [91].

The *S. pombe* Lsd1 protein structure has not been resolved but is conserved to its human homolog LSD1/KDM1a (43% sequence alignment identity). We first performed a computational domain structural analysis to figure out which part of the mammalian LSD1 protein shares similarities with the HMG-box proteins. The overlapping results of the previously reported crystal structure of KDM1a (PDB ID: 2Z3Y) with a typical HMG-box protein (HMGB1)(PDB ID: 1NHM) in humans suggests that the HMG domain of HMGB1 structurally mimics that of the mammalian LSD1 Tower domain (Fig 1B). Since part of the Tower domain of LSD1 contributes to LSD1/CoREST binding to nucleosomes [93], this finding supports the hypothesis that the HMG domain may be essential for Lsd1 binding to chromatin. To confirm this hypothesis, we performed chromatin-immunoprecipitation combined with sequencing (ChIP-Seq) analysis of Lsd1 and Lsd2 in wild-type and their C-terminal domain mutations (*lsd1-ΔHMG* and *lsd2ΔC*). IgG Sepharose beads were used to pull down FTP-tagged wild-type Lsd1, Lsd2, and their C-terminal domain mutants.

Previous studies have shown that Lsd1/2 proteins bind to the promoters of a few hundred genes [70,73,74], suggesting that Lsd1/2 proteins are selectively recruited to those genes. Our ChIP-Seq analysis yields a highly similar set of genomic loci where Lsd1 and Lsd2 are enriched just upstream of the transcriptional start site (TSS) (Fig 1C). This result indicates that Lsd1 and Lsd2 mostly bind to the promoter region of genes and are likely to cooperate with other transcription factors that are involved in regulating gene expression. While examining pericentromeric and mating-type regions, we observed a notable reduction in heterochromatic silencing in *lsd1* and *lsd2* mutant cells [91]. When comparing wild-type Lsd1 and Lsd2 proteins, we found lower enrichments at these constitutive heterochromatic regions in contrast to their higher enrichments at promoter regions (Figs 1C and S2). However, the enrichments of *lsd1* and *lsd2* C-terminal mutant proteins exhibited a modest increase at specific loci associated with the loss of silencing, in accordance with the reported roles of Lsd proteins in heterochromatic silencing [91]. Notably, the localizations of Lsd1 and Lsd2 are diminished at the promoter regions in the absence of a functional C-terminus (Fig 1C). This result demonstrates that the C-terminal domains of Lsd1 and Lsd2 are involved in their chromatin binding at these regions.

Lsd1/2 complex contains two zinc-finger proteins, Phf1 and Phf2, which may also participate in DNA binding. The loss of the C-terminal domains of the Lsd1/2 proteins might destabilize their association with Phf1 and Phf2, thereby affecting the ability of the complex to bind chromatin. We employed FTP-tagged Lsd1 or Lsd2 alleles and combined the tagged allele with Phf1-myc and Phf2-myc. Using co-immunoprecipitation (Co-IP) followed by western

blotting, we confirmed the protein-protein interactions between Lsd1 and Phf1 (Fig 1D) and Phf2 (Fig 1E). Additionally, we established that the loss of the HMG-domain of Lsd1 does not affect the interaction between Lsd1 and Phf1 (Fig 1D) or Phf2 (Fig 1E), and hence, the loss of the HMG-domain is not sufficient to disrupt the entire complex. We also observed that the loss of the C-terminus of Lsd2 does not disrupt the interactions between Lsd2 and Phf1 (Fig 1F) or Phf2 (Fig 1G). These results suggest that the C-terminus mutants of Lsd1 and Lsd2 do not affect the structural integrity of the Lsd1/2 complex.

Additionally, we tested whether the loss of the C-terminal domains of Lsd1 and Lsd2 would affect the chromatin binding of Phf1 and Phf2. We performed the ChIP assay followed by qPCR for analyzing the localization of Phf1-myc and Phf2-myc at the $sah1^+$ promoter region, one of the most robust binding regions of Lsd1 and Lsd2 in the *S. pombe* genome (S3 Fig), in wild-type, *lsd1-ΔHMG*, and *lsd2-ΔC* cells. Similar to Lsd1 and Lsd2, Phf1 and Phf2 are also enriched at the promoter of $sah1^+$ (Fig 1H). Moreover, these enrichments are diminished when Lsd1 or Lsd2 loses its C-terminal domain (Fig 1H). This result indicates that impaired C-terminal domains of Lsd1 and Lsd2 weaken the chromatin binding of Phf1 and Phf2, further supporting the notion that the HMG domains of Lsd1 and Lsd2 are essential for the chromatin binding of Lsd1/2 complex.

## Lsd1 directly interacts with Phf1 and Phf2

In *S. pombe*, Lsd1 is copurified with Lsd2, Phf1, and Phf2 to form the Lsd1/2, SWIRM1/2, SAPHIRE complex [72,73,75], yet the exact architecture of the complex has not been revealed. To elucidate whether Lsd1 and Lsd2 have direct interactions with Phf1 or Phf2, we employed a budding yeast two-hybrid approach (Fig 1I). The bait proteins are fused with the Gal4 DNA-binding domain (BD), and the prey proteins are fused with the Gal4 activation domain (AD). When the bait protein interacts with the prey protein, the DNA-binding domain and activation domain will be close enough to initiate the reporter gene expression under the control of the Gal4 promoter in *S. cerevisiae*. Our results are based on two reporter genes: ADE2 and MEL1. ADE2 generates Ade2, which encodes the phosphoribosylaminoimidazole carboxylase involved in the *de novo* biosynthesis of purine nucleotides. The expression of ADE2 allows yeast cells to grow on minimal medium lacking adenine (-adenine). MEL1 encodes a secreted enzyme, α-galactosidase, which hydrolyzes the colorless X-alpha-Gal in the medium into a blue end product. The expression of these reporter genes allows cells to grow on -adenine medium or turn blue on a medium containing X-alpha-Gal, indicating the direct interaction between bait and prey proteins. The Lsd2-BD and Phf1-BD strains could self-activate the reporter genes without Gal4 activating domain, suggesting that Lsd2 and Phf1 alone could recruit basal transcriptional factors to initiate reporter gene transcription (S4 Fig). Despite these false positive strains, the remaining bait and prey combinations show that Lsd1-BD/Phf1-AD, Lsd1-BD/Phf2-AD, and Phf2-BD/Phf1-AD colonies turn blue on medium with X-alpha-Gal and grow robustly on medium without adenine (Figs 1I–1K and S4 and S5). These results indicate that Lsd1 directly interacts with Phf1 and Phf2, which also physically associate with each other. However, Lsd2 does not physically interact with Lsd1 or Phf2 (Figs 1L and S5).

## Genetic and physical interactions between Lsd1/2 complex, CLRC, and COMPASS

Since the function of Lsd1 and Lsd2 is to demethylate K4 and K9 on histone H3 [67,73,74], we proceeded to examine the genetic interactions between Lsd1/2 and the sole H3K4 methyltransferase, Set1, and the only H3K9 methyltransferase, Clr4. Strikingly, the genetic crosses and tetrad analyses revealed that *lsd1-ΔHMG set1Δ* and *lsd2-ΔC clr4Δ* double mutants resulted in

inviable daughter cells, suggesting that Lsd1 has essential overlapping functions with Set1, while Lsd2 has critical overlapping roles with Clr4 (Fig 2A). This result also highlights a divergent function between Lsd1 and Lsd2. *lsd1-ΔHMG clr4Δ* cells are viable but sick, which implies a negative genetic interaction between Lsd1 and Clr4 (Figs 2A and S1B). We also recovered *lsd2-ΔC set1Δ* cells, in which *set1Δ* alleviates the growth defect of *lsd2-ΔC* and suggests a positive genetic interaction between Lsd2 and Set1 (Figs 2A and S1C). The triple mutants *lsd1-ΔHMG set1Δ clr4Δ* and *lsd2-ΔC set1Δ clr4Δ* are lethal (S1D Fig). Since mutations in all three copies of histone H3, resulting in H3K4A or H3K9A mutants, do not induce lethality [48], our genetic result indicates that Set1, Clr4, and Lsd1/2 likely possess additional functions beyond histone modification that are essential for cell survival. We also noticed *lsd2-ΔC* recues *set1Δ* silencing defects at the mating type locus (S1E Fig). This result is consistent with the opposite roles of Lsd2 and Set1 in dynamically modifying histone H3K4. To summarize, these results reveal the roles of Lsd1 and Lsd2 are differentially associated with Clr4 and Set1.

The unexpected genetic interactions between Lsd1/2 with Clr4 and Set1 prompted us to further investigate the functional connection between these histone-modifying enzymes. Set1 is the catalytic subunit of COMPASS, which interacts with RNA Polymerase II through its subunit Swd2 [94,95]. Swd2 has also been shown to interact with transcription termination machinery through the cleavage and polyadenylation factor (CPF) independently of its functions in H3K4 methylation as part of the COMPASS complex [94,96]. Interestingly, Swd2 is the only protein in the complex that is essential for cell viability in budding yeast [95]. COMPASS also includes the WD40-domain proteins Swd1/Swd3 [97], the tri-methylation specific factor Spp1 [98], the yeast-specific subunit Shg1, the Lid2-demethylase complex subunit Sdc1, and the PHD finger domain protein Ash2 (Fig 2B) [99]. Clr4 forms the CLRC complex along with other members, including the E3 ubiquitin ligase Cul4, the β-propeller protein Rik1, and the Rik1-associated factors (RAF) Raf1 and Raf2 (Fig 2B) [62,100]. Previous pair-wise protein-protein interaction studies placed Raf2 at the center of the CLRC complex with confirmed direct interactions with most other complex members [101]. To provide more direct evidence about the physical interactions between Lsd1/2 complex, COMPASS, and CLRC, we performed co-immunoprecipitation followed by western blotting using GFP-HA-tagged Swd2 (COMPASS) and myc-tagged Raf2 (CLRC) at their C-terminus. Both Lsd1 and Lsd2 interact with Swd2 (Fig 2C and 2E) and Raf2 (Fig 2D and 2F). Taken together, we identified genetic and physical interactions between Lsd1/2 complex, CLRC, and COMPASS, suggesting a cross-talk between these histone-modifying enzyme complexes (Fig 2B).

## Set1 and Clr4 oppositely regulate the genomic binding levels of Lsd1 and Lsd2

The genetic and physical interactions made us wonder whether the loss of COMPASS or CLRC would affect the genomic localization of Lsd1 or Lsd2. We performed ChIP-Seq analysis of Lsd1 and Lsd2 in wild-type, *clr4Δ*, and *set1Δ* cells (Fig 2G and 2H). We averaged the same genes and genomic regions as those shown in Fig 1C. Loss of Clr4 or Set1 does not affect the overall pattern of genomic localizations of Lsd1 and Lsd2. To our surprise, *clr4Δ* increases the enrichments of Lsd1/2 while *set1Δ* decreases the enrichments of Lsd1 at its enriched genomic loci (Figs 2G, 2H, S6 and S7). This finding suggests that *clr4Δ* increases, while *set1Δ* decreases, the genomic binding of Lsd1/2 and potentially affects their protein levels.

## Clr4 and Set1 antagonistically regulate the protein levels of Lsd1/2

Next, we investigated the protein levels of Lsd1 and Lsd2 in the absence of Clr4 or Set1. We combined the Lsd1/2-FTP with *clr4Δ*, *set1Δ*, or both. Our western blot result demonstrates

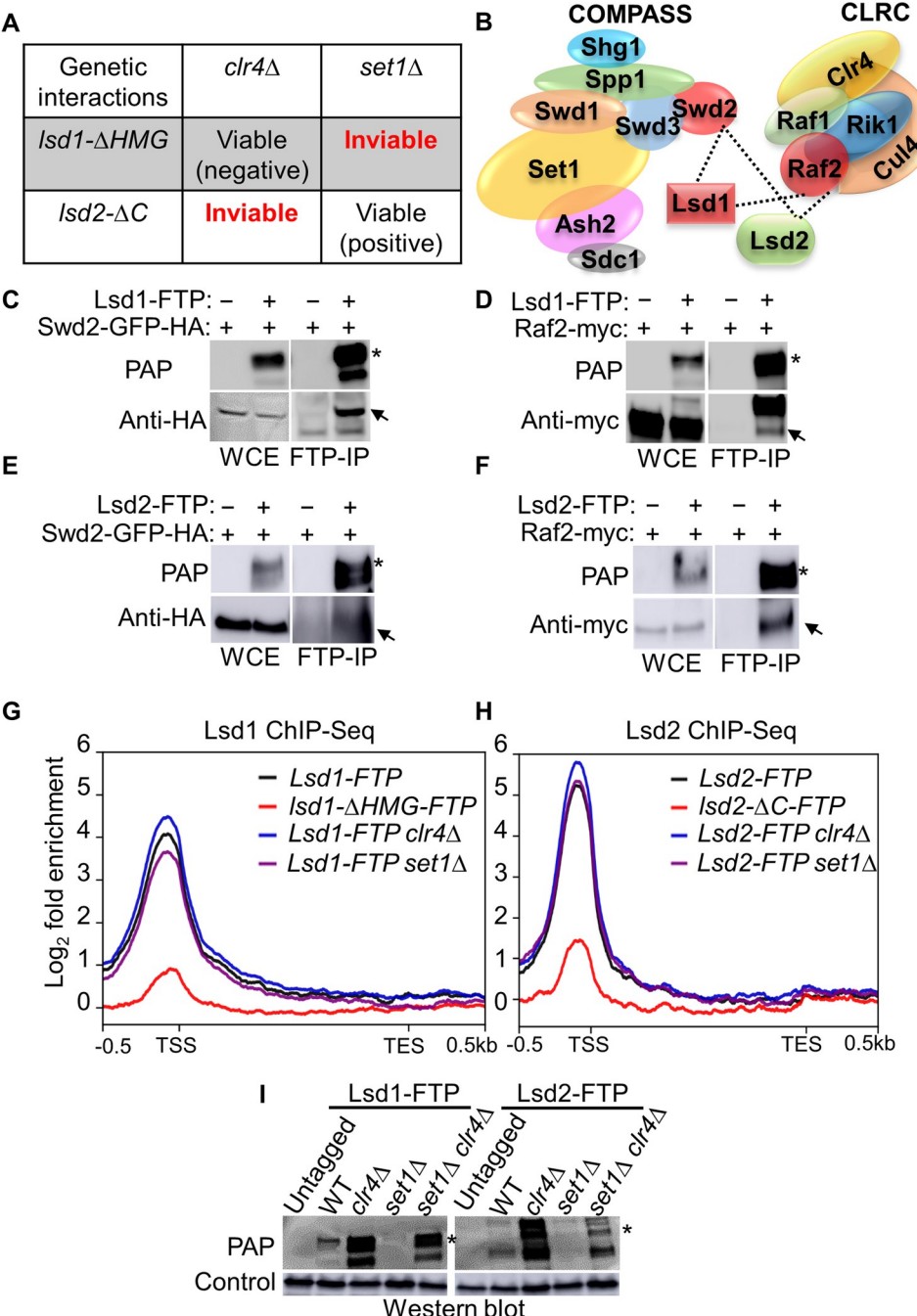

**Fig 2. Genetic and physical interactions between Lsd1/2, CLRC, and COMPASS.** (A) The genetic interactions between *lsd1-ΔHMG* or *lsd2ΔC* with the loss of the methyltransferases Clr4 and Set1. (B) A schematic illustration of the key members of the CLRC and COMPASS complexes, highlighting their functional interactions with Lsd1/2. The dashed lines indicate the protein interactions that were detected by Western blot. (C-F) Protein-protein interactions between Lsd1 and Swd2 (C) or Raf2 (D), and between Lsd2 and Swd2 (E) or Raf2 (F) were revealed through co-immunoprecipitation followed by western blotting. +/- signs indicate the presence or absence of the indicated genetic background. IgG Sepharose beads were used for protein immunoprecipitation. WCE: whole cell extract. PAP was used to detect FTP signals. Asterisks mark the expected size for Lsd1-FTP (C-F), while arrows indicate the expected sizes for Swd2-GFP-HA (C & E), Raf2-myc (D & F). (G-H) ChIP-Seq peak enrichments for Lsd1 (G) and Lsd2 (H) with indicated genetic background are mapped to genomic regions spanning 500 base pairs upstream of TSS to 500 base pairs downstream of the TES. Aligned Lsd1 or Lsd2 occupancy values for all selected genes are plotted as moving averages along their relative genomic positions. The gene sets and genomic regions correspond to those depicted in Fig 1C. Color bars represent different genetic backgrounds, and Log₂ fold enrichment indicates the Log₂ fold change of

Lsd1 and Lsd2 occupancy at specific genome loci compared to the untagged negative control. (I) Analysis of WCE from cells expressing untagged proteins and FTP-tagged Lsd1 and Lsd2 through SDS-PAGE and western blotting. PAP was used to detect Lsd1/2 protein levels. "Untagged" denotes cells with no tagged protein, while "WT" indicates cells expressing wild-type Lsd1/2-FTP. Mlo3 levels were used as the loading control. Asterisks signify the expected full-length size for Lsd1-FTP and Lsd2-FTP. The denotations for "Untagged" and "WT" remain consistent in the rest of the figures.

that without Clr4, both Lsd1-FTP and Lsd2-FTP show an increase in protein levels, compared to the wild-type cells (Fig 2I). In contrast, the protein levels of Lsd1-FTP and Lsd2-FTP show a notable decrease in the absence of Set1 (Fig 2I). Additional examination of GFP-tagged Lsd1 and Lsd2 protein abundance in cells using confocal microscopy confirms that Clr4 and Set1 deletion differentially affects Lsd1- and Lsd2-GFP levels (S8 Fig). Moreover, strain backgrounds lacking both Clr4 and Set1 show intermediate protein levels of Lsd1 and Lsd2 (Fig 2I). Consequently, we wondered whether this effect is initiated at the transcription level. Using RNA-seq and qRT-PCR, we investigated the mRNA levels of Lsd1 and Lsd2 in the absence of Clr4 or Set1 (S9 Fig). Lsd1 and Lsd2 mRNA levels do not correspond with alterations in protein levels of Lsd1 and Lsd2 without Set1 or Clr4. To sum up, it is likely that Clr4 and Set1 regulate the protein abundance of Lsd1 and Lsd2 through a post-transcriptional mechanism.

Deletion of specific components of COMPASS affects the integrity of the complex and stability of Set1 [31,48,102–105]. The deletion of Spp1, Swd1, Swd2, and Swd3 significantly lowers the protein levels of Set1 [48]. To test whether the decreased Lsd1/2 proteins are due to the loss of specific members of COMPASS, we generated independent deletion mutations of each member of the complex (*set1Δ*, *spp1Δ*, *swd1Δ*, *swd3Δ*, *swd2Δ*, *ash2Δ*, *shg1Δ*, *sdc1Δ*) and combined those deletions with Lsd1-FTP and Lsd2-FTP through genetic crossing. For Lsd1, our result shows that the loss of most members of the COMPASS complex, including Set1, Spp1, Swd1, Swd3, Swd2, and Ash2 leads to a significantly decreased amount of Lsd1 protein (Fig 3A). However, the loss of Shg1 and Sdc1 seems to have little or no effect on Lsd1 protein levels. Our result indicates that COMPASS components that alter the protein levels of Set1 also lower the protein level of Lsd1 proteins [48]. A similar pattern was observed in the Lsd2 samples, although the Lsd2 protein levels also decreased with *shg1Δ* (Fig 3B).

Similarly, we generated independent deletions of each member of the CLRC complex (*clr4Δ*, *raf1Δ*, *rik1Δ*, *raf2Δ*, and *cul4-1*, which is a *cul4* mutant allele previously shown to affect Cul4 functions through a reporter marker disrupting the 3′-UTR region of *cul4⁺*) [100] and combined them with Lsd1/2-FTP. The western blot results suggest that the loss of any member of CLRC complex enhances the protein amount of Lsd1 (Fig 3C) or Lsd2 (Fig 3D). Therefore, the integrity of the CLRC complex is required to restrict the protein levels of Lsd1 and Lsd2. Cul4 is the E3 ubiquitin ligase in the CLRC complex [100]. Cul4 also belongs to Cul4-Ddb1$^{Cdt2}$ E3 ubiquitin ligase complex (CRL4 complex) [83,84]. Thus, we wondered whether Cul4 might also regulate Lsd1/2 protein levels through the Cul4-Ddb1$^{Cdt2}$ complex. We investigated the protein levels of Lsd1 and Lsd2 without Ddb1. Loss of Ddb1 slightly enhances the Lsd1-FTP level, while having no significant effect on the Lsd2-FTP level (Fig 3E). It is likely that both CLRC and CRL4 complexes regulate Lsd1 protein levels, although CLRC may play a dominant role. Together, our results show that Clr4 restricts the protein levels of Lsd1 and Lsd2, while Set1 promotes them.

## The ubiquitin-proteasome system (UPS) controls the protein amount of Lsd1 and Lsd2

In mammals, the LSD1 protein level is balanced by ubiquitination and deubiquitination [106–108]. The lysine or arginine residues of LSD1 could be the target sites for ubiquitination

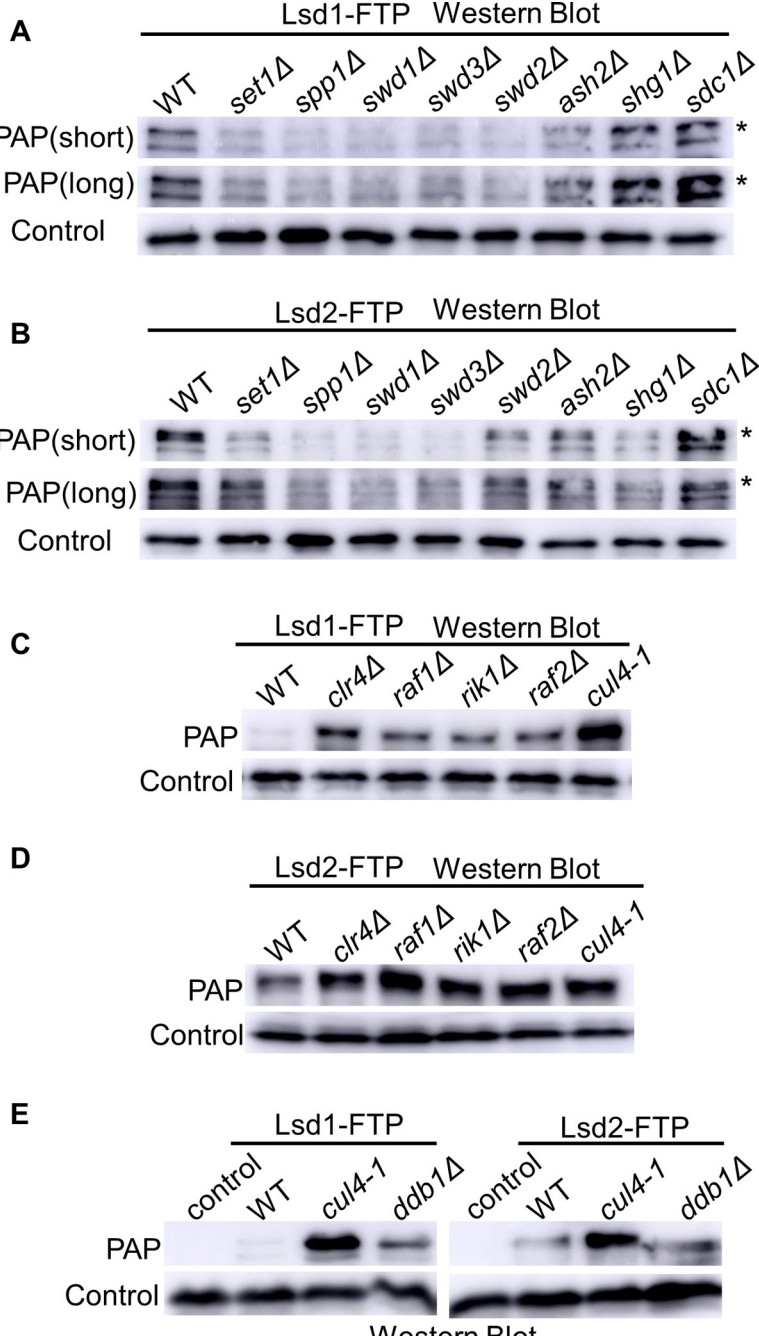

**Fig 3. Clr4 and Set1 antagonistically regulate protein levels of Lsd1/2 proteins.** (A-D) Lsd1-FTP (A & C) and Lsd2-FTP (B & D) protein abundance were analyzed by SDS-PAGE and western blot in the loss of indicated COMPASS complex members (A-B) or CLRC complex members and the *cul4* mutant (*cul4-1*) (C-D). Asterisks denote the expected full-length size of Lsd1-FTP (A) and Lsd2-FTP (B). (E) Examination of Lsd1-FTP and Lsd2-FTP protein levels via SDS-PAGE and western blotting in the wild-type, *cul4-1*, and *ddb1Δ* backgrounds. PAP was utilized to detect Lsd1/2 protein levels. Mlo3 levels were employed as the loading control.

[109,110]. Knowing this, we asked whether Lsd1 and Lsd2 in fission yeast might be degraded by UPS, similar to mammalian LSD1. We conducted a ubiquitination assay followed by pull-down Lsd1-FTP and Lsd2-FTP combined with western blotting. Elevated ubiquitination of

Lsd1 (Fig 4A) or Lsd2 (Fig 4B) was detected in the presence of a temperature-sensitive 26S proteasome subunit mutant, *mts2-1*, which inactivates the proteasome at 33˚C [111] and thereby stabilizes Lsd1 and Lsd2 (Fig 4A and 4B). These results provide evidence that Lsd1 and Lsd2 are degraded through the UPS.

## Set1 contributes to heat-induced upregulation of Lsd1/2 proteins

Clr4 and its associated heterochromatin factors are recruited to facultative heterochromatin domains to modulate a genome-wide transcriptional response to suboptimal environmental conditions [112–114]. Set1 plays a major role in ribosomal gene repression during the cellular response to environmental stressors [115] and cooperates with other factors (including CENP-B, and the HDACs: Clr3 and Clr6) to mediate the transcriptional activation of a certain subset of stress-response genes [116]. We next wondered whether Lsd1/2 protein levels are modulated under stress conditions in a Set1- or Clr4-dependent manner.

In *S. pombe*, heat stress (higher than the optimal range of 28–32˚C) is extensively studied due to its highly conserved heat stress regulatory pathways, which resemble those found in higher eukaryotes [117]. We shifted the cell culture temperature from permissive 30˚C to heat stress conditions (37˚C) for one generation time (2 hours). Cells growing consistently at 30˚C served as the control. Both Lsd1 (Fig 4C) and Lsd2 (Fig 4D) protein levels drastically increase after heat treatment, although the mRNA levels of Lsd1 and Lsd2 were not significantly altered between wild-type and *set1Δ* cells (S10 Fig), suggesting that enhanced Lsd1/2 proteins are required for cell survival under heat stress [61]. Protein levels of Lsd1 and Lsd2 are similar between wild-type and *clr4Δ* cells at 37˚C, indicating that Clr4 is not responsible for Lsd1/2 upregulation in this condition (Fig 4C and 4D). However, without Set1, Lsd1/2 protein levels are no longer upregulated during heat stress (Fig 4C and 4D), which implies that Set1 is required to elevate the protein levels of Lsd1 and Lsd2 under heat stress. Indeed, *set1Δ* cells show severe growth defects at 37˚C [47], which is consistent with previously elucidated roles of Set1 under heat stress.

We next investigated whether Set1 protects Lsd1 and Lsd2 from ubiquitin-mediated protein degradation, which may be more pronounced at 37˚C. Loss of Set1 does not further elevate Lsd1 protein nor ubiquitination levels in *mts2-1*, suggesting that the Set1-dependent degradation of Lsd1 is dependent on the UPS (Fig 4E). Under heat stress (37˚C), it was consistently found that ubiquitination of Lsd1 and Lsd2 is drastically enhanced in *set1Δ* cells, even without *mts2-1* as a background (Fig 4F). In contrast, we only observed a slight decrease in ubiquitination of Lsd1 and Lsd2 in the *clr4Δ* background (Fig 4F), suggesting that Clr4 likely further destabilizes Lsd1 and Lsd2 at 37˚C. In brief, our data indicate that Set1 may protect Lsd1/2 proteins from degradation under heat stress, while Clr4 may participate in their degradation.

## Set1-dependent upregulation of Lsd1/2 proteins is crucial for regulating gene expression under heat stress

In human cells, overexpression of LSD1 is involved in the proliferation, inhibition of apoptosis, and metastasis of several types of cancers, such as gastric, breast, and prostate cancers [109, 118, 119]. The elevated levels of Lsd1/2 proteins at 37˚C suggest that their functions might be essential for cell survival at stressful high temperatures. Indeed, compared to the wild-type cells, the expression of numerous genes is altered in *lsd1-ΔHMG* and *lsd2-ΔC* mutants at 37˚C, and this alteration is correlated with that loss of Set1 (Fig 4G and S1 Table). Intriguingly, about 72% of downregulated genes in *lsd1-ΔHMG*, *lsd2-ΔC*, and *set1Δ* are antisense non-coding RNAs (S1 Table), indicating that Set1-mediated elevation of Lsd1/2 protein functions to stimulate antisense transcription under heat stress. The gene expression pattern alteration showed

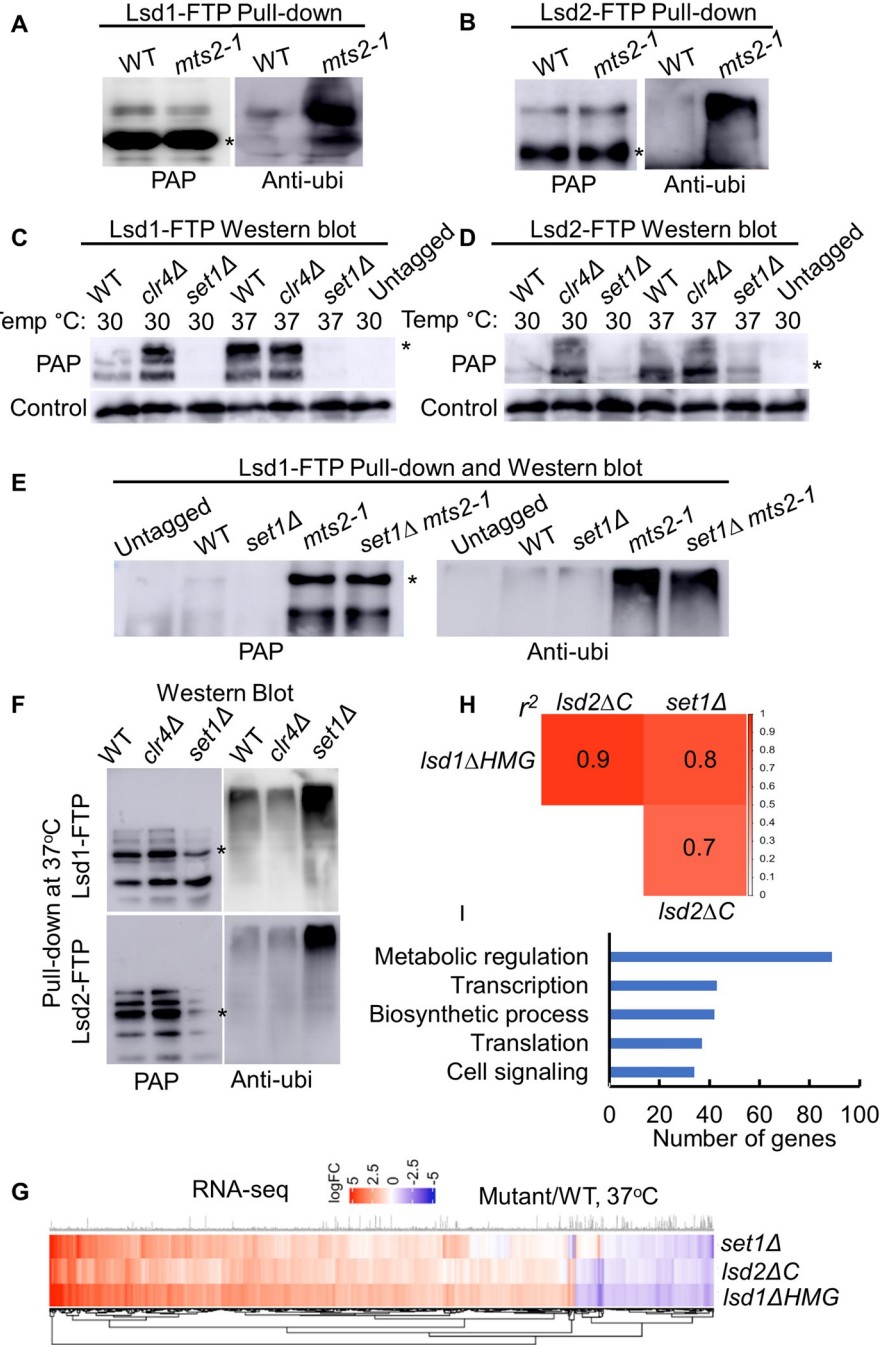

**Fig 4. Set1-dependent upregulation of Lsd1/2 proteins is crucial for regulating gene expression under heat stress.**
(A-B) Assessment of Lsd1-FTP (A) and Lsd2-FTP (B) ubiquitination levels in wild-type and *mts2-1* strains. Cells were cultured in a rich medium (YEA) at 33°C for 8 hours. (C-D) FTP-tagged Lsd1 (C) and Lsd2 (D) protein abundance at indicated genotypes were analyzed by SDS-PAGE and western blot. The cells were cultured at the permissive temperature (30°C) and heat stress conditions (37°C) for 2 hours after the initial culture at 30°C. (E) Analysis of Lsd1-FTP ubiquitination levels in indicated strains. Cells were cultured in a rich medium (YEA) at 33°C for 8 hours. (F) Evaluation of Lsd1-FTP and Lsd2-FTP ubiquitination levels in indicated strains. Cells were cultured at 30°C in a rich medium and then subjected to a temperature shift to 37°C for 2 hours. (A, B, E, and F) IgG Sepharose beads were used to precipitate Lsd1-FTP or Lsd2-FTP, followed by western blotting using specified antibodies. PAP was employed to detect Lsd1/2 protein levels, and an anti-ubiquitin antibody was used to detect ubiquitin signals. Asterisks indicate the expected size for full-length Lsd1-FTP and Lsd2-FTP. (G) Hierarchical clustering of *set1Δ*, *lsd1ΔHMG*, and *lsd2ΔC* mutants at 37°C based on the similarities in their expression profiles. Gene expression was compared between mutants and wild-type under 37°C conditions using RNA-Seq data. The color scale ranges from blue to red, reflecting

increasing fold expression compared to the control. (H) Pearson's correlation coefficients were translated into color codes, illustrating similarities in gene expression pattern alterations across different genetic backgrounds. The numbers represent the degree of similarity in gene expression patterns. (I) Gene Ontology analysis of overlapping genes based on the functions of their products. The top five terms with the most assigned genes are presented. These terms encompass genes involved in metabolic regulation (89 genes), transcription (43 genes), biosynthetic processes (42 genes), translation (37 genes), and cell signaling during stress (34 genes).

high similarity between *lsd1-ΔHMG*, *lsd2-ΔC*, and *set1Δ* during heat stress. At least 70% of genes share similar regulation patterns (between *lsd2-ΔC* and *set1Δ*) and the highest similarity can reach 90% (between *lsd1-ΔHMG* and *lsd2-ΔC*) (Fig 4H). We further analyzed and classified those differentially expressed genes according to their functional groups. Those genes participate in metabolic pathways, transcription, biosynthesis, translation, and cell signaling, which are closely related to the stress response pathways (Fig 4I and S2 Table). To conclude, Set1-dependent upregulation of Lsd1/2 proteins is critical for controlling gene expression under heat stress.

## CLRC controls Set1 protein level

In budding yeast, Set1 protein levels are reduced in mutants deficient in Swd1, Swd2, Swd3, or Spp1 [102,104,120]. In *S. pombe*, Set1 protein amount is differentially affected without COMPASS subunits. Set1 proteins were barely detectable in *swd1Δ* and *swd3Δ* and were noticeably reduced in *swd2Δ* and *spp1Δ* mutants, but were minimally affected by the loss of Ash2, Sdc1, and Shg1 [48]. Since Cul4 is an E3-ubiquitin ligase, we next investigated whether CLRC may regulate Set1 protein level by controlling Set1 degradation through the UPS, thereby modulating the protein levels of Lsd1/2. We, therefore, visualized the levels of fully functional N-terminal Flag-tagged Set1 (Flag-Set1) in wild-type or CLRC mutant cells [48]. We found that the loss of any members in CLRC promotes the protein levels of Set1, both at 30°C or 37°C (Fig 5A), indicating that the intact CLRC restricts the amount of Set1. Next, we investigated whether chromatin-bound Clr4 is necessary for regulating Set1 protein level. The binding of Clr4 to methylated H3K9 is impaired when its chromodomain of Clr4 (*clr4W31G*) is mutated [63]. Although no significant alterations were detected for Flag-Set1 protein levels between wild-type and *clr4W31G* cells at 30°C, the elevated Flag-Set1 is less pronounced in *clr4W31G* compared to *clr4Δ* at 37°C (Fig 5B), indicating that chromatin-bound Clr4 may play a role in regulating Set1 protein level, at least under heat stress. We also investigated the Set1 protein level without Ddb1. Both *cul4-1* or *ddb1Δ* enhanced the amount of Set1 protein, indicating that both CLRC and CRL4 complexes modulate Set1 levels (Fig 5C). *cul4-1* elevates Set1 significantly at 37°C compared to that at 30°C, which is due to the temperature sensitive nature of *cul4-1*. It is possible that Cul4 targets Set1 for degradation through the ubiquitin-proteasome system.

## HULC is essential for Set1-dependent upregulation of Lsd1/2 proteins under heat stress

The ubiquitination on histone H2B (H2B K119ub) is a prerequisite mark for H3K4 methylation [121–123]. In *S. pombe*, it is catalyzed by the histone H2B ubiquitin ligase complex (HULC). Without HULC, no H3K4me occurs [54,124]. In *S. cerevisiae*, Set1 is not detectable in the absence of HULC or H2B K123ub [48,125]. In *S. pombe*, however, the Set1 protein level is only reduced in the absence of H2B ubiquitination [48]. We confirmed this finding by checking the levels of Flag-tagged Set1 protein with the deletion of Brl1 or Brl2 (Fig 6A). Brl1 and Brl2 both belong to HULC, and their activities are essential for H2B K119 ubiquitination

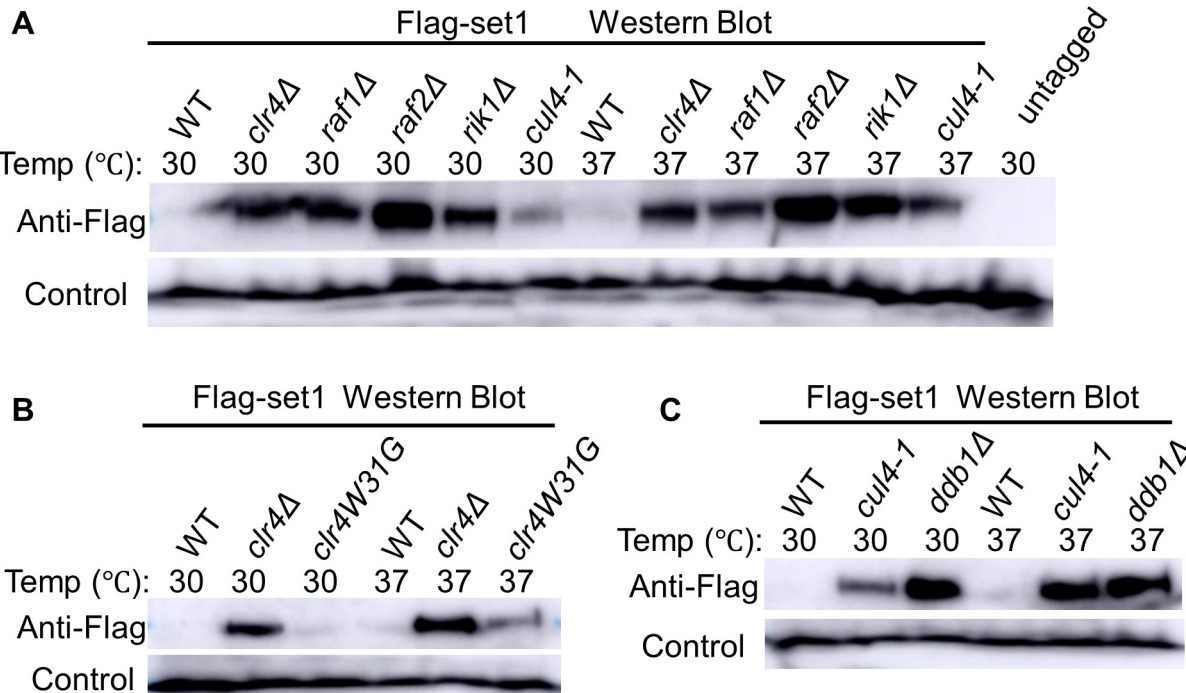

**Fig 5. CLRC controls Set1 protein level.** (A-C) Whole-cell extracts from untagged and Flag-tagged Set1 cells were subjected to SDS-PAGE and western blot using an anti-Flag antibody. The assessed Set1 protein levels in indicated genetic backgrounds at 30°C and 37°C for 2 hours after initial culture at 30°C. Mlo3 levels served as the loading controls.

[54,126,127]. In wild-type cells, we observed a significantly increased amount of Set1 at 37°C (Fig 6A), indicating that Set1 is also stabilized during heat stress. Cells without Brl1 or Brl2 do not show a similar increased level of Set1 at 37°C, suggesting that H2B ubiquitination is required for stabilizing Set1 during heat stress (Fig 6A). We also investigated how H2B ubiquitination influences Lsd1/2 proteins. During heat stress, the loss of Brl1 and Brl2 reduces the abundance of Lsd1 (Fig 6B) and Lsd2 (Fig 6C) proteins, which is likely caused by the downregulation of Set1. To figure out whether Lsd1 and Lsd2 are affected by particular histone modifications, especially H3K4me and H2B K119ub, we analyzed Lsd1/2 protein levels in H3K4R and H2B K119R mutants, which are no longer subject to their covalent modifications at these specific sites. At 37°C, only marginal changes in Lsd1 protein levels were observed in H3K4R and H2B-K119R mutants (Fig 6D). However, a significant reduction of Lsd2 was seen in both H3K4R and H2B K119R mutants (Fig 6E), which is consistent with the synergistic role of Lsd2 with H3K4 methylation. The downregulation of Lsd1/2 proteins in H2B K119R mutants is consistent with our observations in *brl1Δ* or *brl2Δ* cells. Overall, we conclude that HULC is required for Set1-dependent upregulation of Lsd1/2 proteins under heat stress.

## Discussion

In recent studies, the functions of Lsd1 and Lsd2 histone demethylases (Lsd1/2) have extended beyond their demethylation catalysis and regulatory pathways [91]. In this study, we unveil a network of cross-regulation, in which two histone lysine methyltransferases with opposing roles in gene expression, Set1 and Clr4, antagonistically control the protein levels of Lsd1/2. While counterintuitive at first glance, the elevation of Lsd protein levels by Set1 could lead to an amplification of Lsd protein activities, which in turn counteract the function of Clr4 and

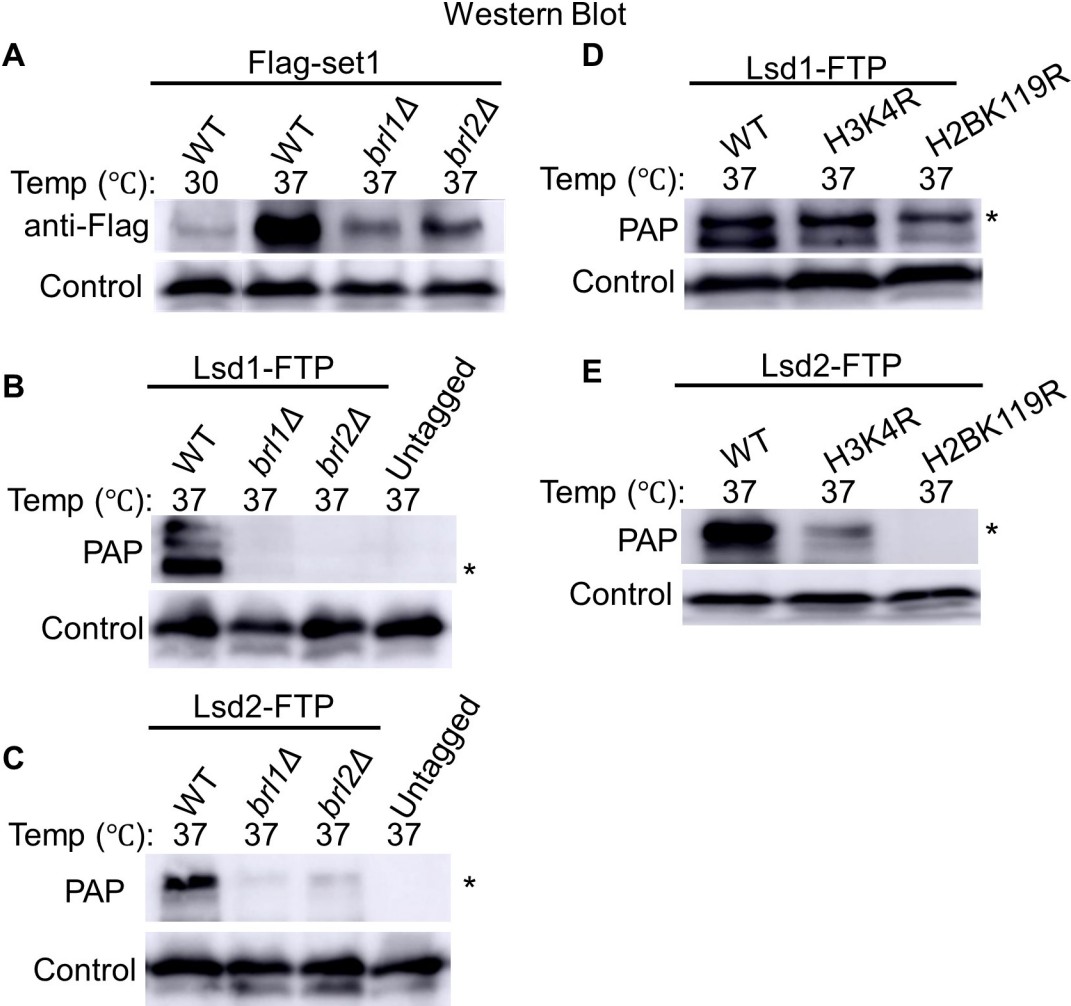

**Fig 6. HULC is essential for Set1-dependent upregulation of Lsd1/2 proteins under heat stress.** (A) Flag-tagged Set1 levels were assessed via SDS-PAGE and western blot in *brl1Δ* and *brl2Δ* strains during 2-hour heat stress (37°C) after initial culture at 30°C. (B-C) Protein levels of Lsd1-FTP (B) and Lsd2-FTP (C) were examined in *brl1Δ* and *brl2Δ* strains under the same heat stress condition. (D-E) Lsd1-FTP (D) and Lsd2-FTP (E) protein levels were evaluated in H3K4R and H2BK119R backgrounds during a 2-hour heat stress at 37°C. Mlo3 levels were used as loading controls. Asterisks denote the expected full-length size of Lsd1-FTP (B & D) and Lsd2-FTP (C & E).

prevent excessive Clr4 activity. This intricate cross-regulation among these enzymes ensures fine-tuned activities, creating a flexible regulatory circuit that upholds cellular homeostasis. These findings broaden the regulatory landscape of the CLRC, COMPASS, and Lsd1/2 complexes, which are vital and highly conserved protein-modifying complexes with diverse roles in genome control and development.

## The C-terminal domains of Lsd1 and Lsd2

Our previous studies have shown that the C-terminals of Lsd1 and Lsd2 are required for gene silencing at heterochromatic regions [91]. The combined absence of RNAi machinery, histone deacetylases (HDACs), and the *lsd1-ΔHMG* and *lsd2-ΔC* mutants intensify the impairments in gene silencing across all constitutive heterochromatin regions [91]. In mammals, LSD1 interacts with different transcriptional repressors, such as Co-REST and HDAC complexes, to

promote gene silencing and enhance the repressive chromatin status [128–130]. The TOWER domain offers a binding site for LSD1 to interact with other protein complexes, including Co-REST and HDACs [131–133]. However, in fission yeast, Lsd1 does not possess a TOWER domain. Nevertheless, both the TOWER domain of LSD1 and the HMG domain of Lsd1 exhibit similar DNA-binding activities [131,134]. Our ChIP-Seq data confirmed that both C-terminals of Lsd1 and Lsd2 are involved in chromatin binding, which may explain the gene silencing defect that occurs when the C-terminals of Lsd1 and Lsd2 are lost. While we were surprised to find that the loss of the C-terminal regions does not impact the integrity of the complex (Fig 1D–1G), it does lead to a deficiency in the chromatin binding of Phf1 and Phf2 (Fig 1H). These findings indicate that the HMG domain may not completely overlap in function with the TOWER domain. Phf1 and Phf2 are PHD-domain-containing proteins and are thought to play a role in targeting the Lsd1/2 complex to heterochromatin boundaries and some gene promoter regions or modulating the catalytic specificity of Lsd1 and Lsd2 [76]. Thus, our results suggest that in *S. pombe*, Lsd1/2 proteins initially bind to the chromatin, and, subsequently, recruit Phf1 and Phf2 to form the complex, indicating that Phf1 and Phf2 could be further stabilizing the localization of Lsd1 and Lsd2 on chromatin or may function as an adaptor that interacts with additional protein complexes in the same regulatory pathway.

By employing a yeast two-hybrid approach, we established the direct physical interactions between Lsd1, Phf1, and Phf2, thereby providing evidence for the formation of a central protein core. We could not reveal direct interactions between Lsd2 and the other Lsd1/2 complex components (Figs 1J–1K, S4 and S5). In humans, LSD1 and LSD2 possess distinct structures that allow them to associate with distinct protein complexes and localize at different genomic loci [134]. Compared to LSD1, LSD2 lacks the TOWER domain and, therefore, cannot form the stable complex with Co-REST [131,135,136]. In mammals, LSD2 nucleosome-demethylase activity is dependent on a distinct linker peptide originating from the multidomain protein NPAC (a cytokine-like nuclear factor, also named GLYR1 or NP60) [70,137], which has no ortholog in *S. pombe*. Although Lsd2 is copurified with Lsd1, Phf1, and Phf2 [72,73,75], it is plausible that certain unidentified proteins in *S. pombe* act as adaptors, similar to NPAC, facilitating the assembly of the Lsd1/2 complex.

## The genetic interactions between Set1, Clr4, and Lsd1/2

Genetic interactions offer a valuable approach to investigate the functional connection between gene products. A negative genetic interaction suggests that two genes operate in distinct parallel pathways or lack functional association, while a positive genetic interaction suggests that two genes function either oppositely or in the same pathway [138]. Surprisingly, *lsd2-ΔC* exhibited synthetic lethality when combined with *clr4Δ* but not *set1Δ*, whereas *lsd1-ΔHMG* resulted in synthetic lethality when combined with *set1Δ* but not *clr4Δ* (Fig 2A). Thus, we proposed that Lsd1 operates in a similar pathway to Clr4, while Lsd2 functions in a related pathway to Set1. Indeed, we revealed a positive genetic interaction between *lsd2-ΔC* and *set1Δ* (S1C Fig), aligning with their contrasting roles in modifying H3K4. Additionally, our investigation revealed a physical interaction between Lsd1/2, COMPASS, and CLRC components (Fig 2B–2F), underscoring the significant functional and physical associations between these enzymes (Fig 2A–2F). Moreover, we employed a comprehensive genome-wide approach to examine the involvement of Clr4 and Set1 in regulating Lsd1/2 localization at promoter regions, which shows that although the chromatin binding patterns of Lsd1/2 remain unaffected in *set1Δ* and *clr4Δ* strains, these enzymes do exert control over the binding levels of Lsd1/2 (Figs 2G, 2H, S6 and S7). These findings suggest that Set1 facilitates the accumulation of Lsd1/2 protein, while Clr4 serves to restrict the quantity of Lsd1/2 present.

## Clr4 and Set1 might directly modify Lsd1/2 to modulate their protein levels

While Set1 and Clr4 are known to have crucial roles in transcriptional regulation, our RNA-seq and qRT-PCR analysis revealed that the altered mRNA levels of Lsd1 and Lsd2 are not correlated with the loss of Set1 or Clr4 (S9 Fig). Instead, it appears that Set1 and Clr4 influence Lsd1/2 protein levels through post-transcriptional mechanisms (Figs 2I, S8 and 4). Interestingly, when various components of the COMPASS complex were deleted, we observed a decrease in the abundance of Lsd1 with the exception of Shg1 and Sdc1, and Lsd2 except for Sdc1 (Fig 3A and 3B). Set1 is known to be an unstable protein in budding yeast because the H3K4me level highly affects Set1 stability [125]. Nonetheless, in fission yeast, Set1 is more stable; however, the integrity of the COMPASS complex is still important for maintaining its stability [48]. These findings align with our observation that disruption of COMPASS results in reduced levels of Set1, thereby compromising its stability, and further modulating the levels of Lsd1/2 protein. One interesting result comes from the decreasing level of Lsd2 without Shg1. It has been reported that Shg1 interacts with a Ran-binding protein, Mog1, which is required for H2B ubiquitination and the maintenance of H3K4me3 levels in budding yeast [139]. The loss of Shg1 may impair the function of Mog1, resulting in an H2B ubiquitination defect. This is consistent with our result that Lsd2 is more sensitive to blocked H3K4 methylation and H2B K119 ubiquitination (Fig 6E). Since Lsd2 and Set1 have a positive genetic interaction, Lsd2 may participate in a H2B K119ub-mediated Set1 regulation pathway. While Shg1 is not important for the assembly and integrity of the COMPASS complex, it is the only subunit that restrains H3K4me3 in yeast cells [97,102]. In humans, LSD1 accumulates at DNA double-strand break (DSB) sites, interacts with the E3 ubiquitin ligase RNF168, and further promotes demethylation of H3K4me2 [140,141]. Moreover, H3K9me3 offers binding sites for DNA damage repair machinery suggesting that one plausible function of LSD1 may be the repression of homologous recombination repair [141–143]. Based on these findings, Lsd2 might participate in DNA damage repair pathways like LSD1 in humans. The depletion of another component of the COMPASS complex, Ash2, does not alter the protein level of Set1, but impairs global H3K4me3 and moderately affects H3K4me2 [48]. We noticed a moderate decrease in the protein levels of Lsd1 and Lsd2 in *ash2Δ*, which might be due to the negative feedback regulation to maintain the proper level of H3K4 methylation, at least di-methylation (Fig 3A and 3B).

The substrates of histone methyltransferases or histone demethylases extend beyond histones themselves. For instance, Clr4 methylates specific residues, K165 and K167, within the carboxy-terminal region of Mlo3, a protein involved in RNA export [144]. Clr4 also exhibits auto-methylation on K455 and K472, a mechanism that prevents internal loop formation and inhibits its own enzyme activity [145]. In *S. cerevisiae*, Set1 targets the kinetochore protein Dam1, and proper methylation of Dam1 is crucial for normal chromosome segregation [146]. In humans, the homolog of Set1, SETD1A, plays a role in cancer cell proliferation and tumorigenesis by methylating heat shock protein HSP70 and the nucleocytoplasmic shuttling protein YAP [147]. Moreover, in humans, LSD1/KDM1a can be tri-methylated directly by SUV39H2, the homolog of Clr4, at K322, leading to the stabilization of LSD1/KDM1a by inhibiting its polyubiquitination [148]. Studies have indicated that during heat stress, the SWIRM domain-containing protein OsHDMA701, belonging to the LSD1-type subfamily in plants, is highly upregulated [149]. Together, these findings suggest that Clr4 and Set1 might directly methylate Lsd1/2 to regulate their protein levels, and this regulatory network might also exist in the higher eukaryotes.

## The roles of CLRC and COMPASS in Lsd1/2 degradation via UPS

The UPS serves as the primary pathway for protein degradation [39]. Our research has confirmed that Clr4 promotes Lsd1/2 degradation (Fig 4F). One crucial component within the CLRC complex is Cul4, which interacts with an adaptor protein Ddb1 and the associated factor Cdt2, to assemble the Cul4-Ddb1$^{Cdt2}$ E3 ubiquitin ligase complex (CRL4 complex) [78,83,84]. In conjunction with the other two subunits of CLRC, the Cul4-Rik1-Raf1 complex exhibits a similar structure to Cul4-Ddb1$^{Cdt2}$ and functions as an E3 ubiquitin ligase [41,88,89]. Our findings indicate that the loss of Ddb1 only slightly enhances the Lsd1-FTP level while having no significant effect on the Lsd2-FTP level (Fig 3E). This suggests that Lsd1/2 are the likely targets of the Cul4-Rik1-Raf1 complex. Clr4, the core subunit of CLRC, and Raf2, which is proposed to be positioned at the center of the CLRC complex, exhibit direct interactions with most other members [101]. While we are not certain that CLRC directly affects Lsd1/2 stability, our findings suggest that losing Clr4 or Raf2 disassembles CLRC, which likely impairs the Cul4-Rik1-Raf1 complex, potentially blocking the ubiquitination and subsequent degradation of Lsd1/2. In budding yeast, approximately 43% of protein methylation sites have the potential to be targeted for ubiquitination [46]. Knowing this, we propose that Set1 methylates specific lysine residues on Lsd1/2, thereby safeguarding them from entering the UPS. Our future investigations will be dedicated to identifying the precise lysine residues that may be targeted by both E3 ubiquitin ligases and methyltransferases.

## The roles of Set1 and Lsd1/2 under heat stress

The impact of Set1 on stabilizing Lsd1/2 is particularly pronounced during periods of heat stress (Fig 4). We observed a remarkable increase in the protein levels of Lsd1/2 when cells were subjected to heat stress conditions (Fig 4C and 4D), indicating that the functions of Lsd1/2 are vital for cell survival under these circumstances. However, in the absence of Set1, we noted no upregulation of Lsd1 and Lsd2, which coincided with global changes in gene expression under heat stress conditions (Fig 4G–4I). In budding yeast, researchers have found that COMPASS complex mediates repressed gene expression of select groups of genes [150]. H3K4me3 largely enriches at the 3′ end of the genes, and the observed repressive effect is most likely due to 3′ end antisense transcription [151]. It is consistent with our data that 72% of downregulated genes in *lsd1-ΔHMG*, *lsd2-ΔC*, and *set1Δ* are antisense non-coding RNAs. Lsd1 and Lsd2 also function to repress gene expression [91], which may work coordinately with Set1 to regulate gene expression during heat stress. Set1 itself exhibits upregulation during heat stress (Fig 6A). Previous studies have shown that the protein level of Set1 decreases in the absence of H2B ubiquitination, which is known to impact the stability and activity of chromatin-associated Set1 [48,59]. In our experiments, we clearly observed reduced levels of Set1 during heat stress in the absence of Brl1 and Brl2, both of which are components of the HULC complex for H2B ubiquitination (Fig 6A) [54]. Consequently, the protein levels of Lsd1 and Lsd2 are also downregulated in the *brl1/2Δ* strains during heat stress (Fig 6B and 6C). Disrupting normal H2B K119ub has a similar effect on Set1 and Lsd1/2 protein levels (Fig 6). Previous studies have confirmed that the absence of H2Bub does not influence the integrity of the COMPASS complex in *S. pombe* and still maintains the catalytic activity of Set1 *in vitro* [56]. However, H2Bub is required for the association of the COMPASS complex with actively transcribed chromatin [56]. Monoubiquitinated H2B helps to properly position the COMPASS complex to nucleosomes or may lead to a conformational change of the nucleosomes to enhance COMPASS complex activity [121,152]. These observations suggest that the chromatin-bound fraction of Set1 may act as an upstream regulator of Lsd1 and Lsd2. No significant change in Set1 protein level was observed during prior research in H3K4R mutants at the

permissive growing temperature [48]. Here, we found that at 37˚C, Lsd1 protein levels were barely affected in the H3K4R mutant and moderately decreased in the H2B K119R mutant (Fig 6D). However, an obvious reduction of Lsd2 was seen in the H3 and H2B mutants (Fig 6E), which indicates that the upregulation of Lsd2 during heat stress is dependent on robust H3K4 methylation.

### CLRC complex monitors the protein level of Set1

Additionally, we discovered that the CLRC complex modulates the protein level of Set1, implying that Set1 itself could be a target of the Cul4-Rik1-Raf1 complex (Fig 5A). CLRC may directly regulate the protein levels of Lsd1/2 or indirectly by modulating the protein level of Set1. Furthermore, Set1 could potentially be a substrate by the chromatin-bound CRL4 complex during heat stress (Figs 5B and 6C). A recent study demonstrated that *S. pombe* cells generate N-terminal truncated histone demethylase, Epe1, in response to external stress [153]. Epe1 antagonizes heterochromatin assembly, which may involve the removal of excess H3K9 methylation from euchromatic regions [154,155]. The truncated Epe1 accumulates in the cytoplasm and results in more H3K9 methylation-heterochromatin and repression of gene expression [153]. Heterochromatin Protein 1 (HP1) is a highly conserved reader protein of H3K9me2/3, which binds to heterochromatic regions. Previous studies have shown that the deficiency of interaction between HP1 and SUV39H1 (Clr4 homolog in mammals) leads to the degradation of SUV39H1 through the UPS [156]. Since *clr4W31G* demonstrates impaired chromatin-binding, *S. pombe* cells may redistribute Clr4 in a similar pathway by producing truncated Epe1.

In summary, our proposed model highlights the role of Set1 in safeguarding the stability of Lsd1/2 proteins by countering degradation through the UPS (Fig 7). Set1 potentially exerts this phenomenon by methylating lysine residues on Lsd1 and Lsd2, which in effect would compete with the ubiquitin E3 ligase and prevent ubiquitination on the same or related sites. Moreover, CLRC, either directly or indirectly, regulates the protein levels of Lsd1/2 and Set1, possibly through the Cul4-Rik1-Raf1 E3 ubiquitin ligase complex.

## Materials and methods

### Yeast strains and cultures

*S. pombe* strains used in this study are listed in the S3 Table. All oligonucleotides used in generating the strains are listed in the S4 Table. Fission yeast was cultured under standard conditions at permissive temperature (30˚C) or heat stress conditions (37˚C, 2 hours) in a shaker incubator. Genetic methods and media composition are as previously described [157]. Site-directed mutagenesis for gene deletion and tagging was performed as described previously [158]. Lsd1 and Lsd2 C-terminal truncations were based on the Lsd1-FTP and Lsd2-FTP tagged strains. Strains were generated either by genetic crossing followed by tetrad dissection or through transformation by electroporation. Diploid strains used in the meiotic induction studies were generated by crossing on Edinburgh minimal media (EMM) for 24 hours, followed by the selection of diploids with complementary *ade6*+ alleles (*ade-210/ade-216*) on selective media lacking adenine (aa-Ade) [159].

### Chromatin Immunoprecipitation (ChIP)

ChIP experiments were performed as previously described [91]. Strains were cultured in 100 mL to the exponential growth phase, and protein-DNA crosslinking was performed with paraformaldehyde with slow shaking for 30 minutes. FTP-tagged ChIP was performed using 50 μL

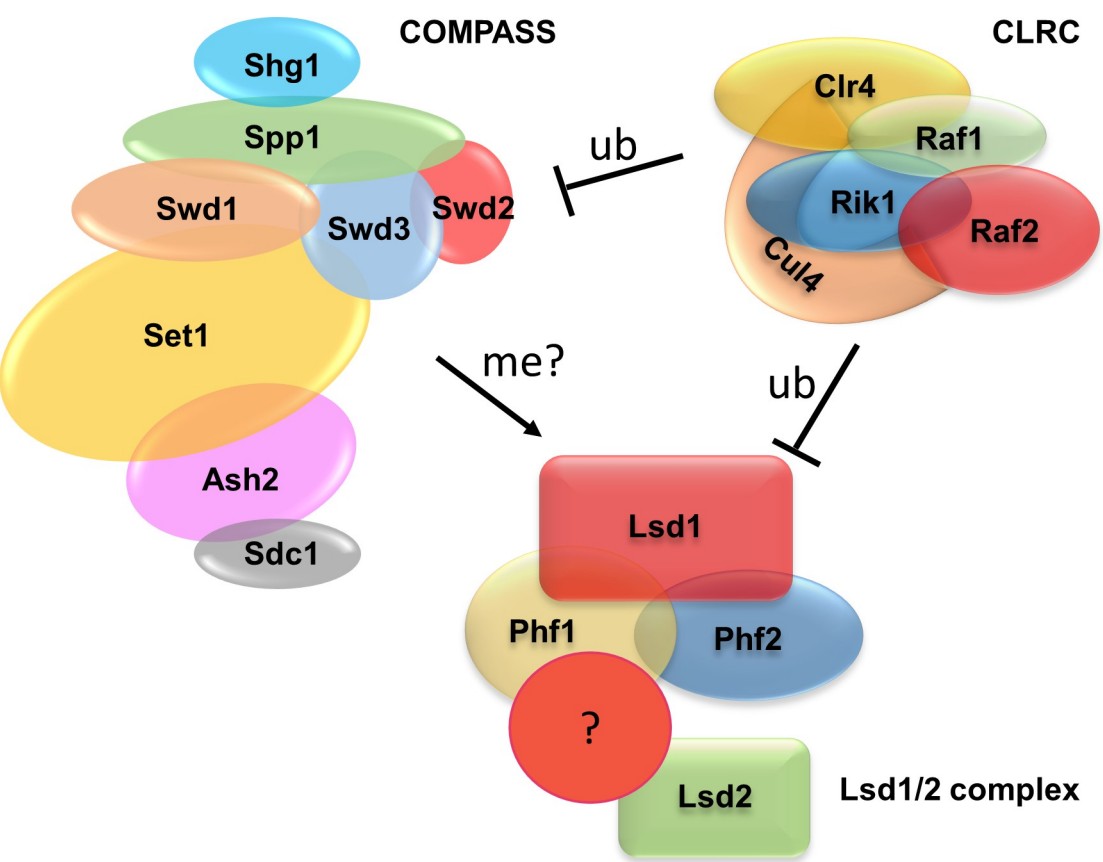

**Fig 7. A model depicting the cross-regulation between CLRC, COMPASS, and Lsd1/2 proteins.** We propose that CLRC functions as a ubiquitin E3 ligase that potentially targets Lsd1/2 or Set1 and promotes their degradation. Set1 safeguards Lsd1/2 from degradation by methylating the specific lysine residues of Lsd1/2, which competes with ubiquitination from CLRC.

IgG Sepharose (GE Healthcare, 17096901) for 4–6 hours at 4°C. myc-ChIPs that were performed in FTP (Protein A-, TEV cleavage site, and Flag- double epitope-tag) backgrounds were modified. After sonication and fragmentation of chromatin, whole cell extracts were incubated with 50 units of AcTEV protease (Invitrogen Corp., 12575015) and 50μL IgG Sepharose (GE Healthcare, 17096901) at 4°C overnight. Pre-cleared whole cell extracts were then collected after centrifugation at 17,530 × g for 10 min, followed by adding the antibody (Anti-myc, Abcam ab9132). Reverse cross-linking and elution were performed at 65°C overnight. ChIP DNA was purified using DNA purification columns (Thermo Fisher, K0512), and quantified by quantitative-PCR (QuantStudio 3, Applied Biosystems) using SYBR Select Master Mix (Applied Biosystems, 4472908). Statistical analysis was performed using a two-tailed Student's $t$ test. Error bars represent the standard error of the mean (s.e.m.).

### RNA-Seq analysis

RNA-Seq analysis was performed as previously described [91]. Additional datasets were extracted from readily available online accession files and included for comparison (GSE148191) [99,112]. RNA was isolated from *S. pombe* strains (each genotype represented by 2 independently derived biological replicates) using the Quick-RNA Miniprep Kit (Zymo Research, R1054) following manufacturer protocols. RNA was recovered in RNase-free water, frozen, and shipped to Novogene where Plant and Animal Eukaryotic Strand Specific mRNA

(WOBI) sequencing was performed with standard bioinformatics service (HiSeq PE150; 150 bp paired-end with directional protocol; 9G raw data per sample). Reads were first trimmed of adapters using Trimmomatic v0.39 under the default settings using adaptors from Tru-Seq2-PE.fa [116]. Each of the 150 bp paired-end RNA-Seq reads for each sample were aligned to the *S. pombe* reference genome ASM294v2 [160] using STAR [161], with an average of $3.7 \times 10^7$ uniquely mapped reads. For the stranded analysis, to select where reads were only captured from a forward or reverse stranded read, files were filtered using SAMtools flags that corresponded to these specifications [162]. Aligned reads were quantified using FeatureCounts [163] with options -t gene -o fraction, which allowed for the quantification of an average of $3.2 \times 10^7$ reads per sample to be assigned to the ASM294v2 genomic feature file. Differential gene expression (DGE) analysis was performed using limma with the—voom extension with Empirical Bayes smoothing of gene-wise standard deviations [164]. An additive model was used where single or double-mutant statuses were treated as covariates [165]. Top DGE genes were those where *P* adjusted $\leq$ 0.0001, and comparisons were made between wild-type and mutants. Further assessment was done by examining fold change differences that either increased or decreased due to either temperature sensitivity or the influence of an additional mutation. Further assessment was done by filtering $\text{Log}_2$ fold-change differences either $\geq$ 2 or $\leq$ -2 to validate current findings to those of previously reported [91]. RNA-Seq data in this study were deposited in the Gene Expression Omnibus with accession number (GSE246595).

## ChIP-Seq analysis

For ChIP-Seq analysis, sequence libraries were taken from >3 ng/μL of ChIP DNA using standard FTP-ChIP protocol as described above [91,166]. ChIP libraries were prepared and sequenced by Novogene using Illumina PE150, Q30$\geq$80%; 150 bp paired-end. Reads were first trimmed of adapters using Trimmomatic v0.39 under the default settings using adaptors from TruSeq2-PE. fa [167]. The DNA reads were then mapped to the *S. pombe* reference genome ASM294v2 55 using bwa mem v0.7.17 [160] with an average of $5.8 \times 10^6$ of reads mapped and paired or an average of 96.9% of all reads mapped to the reference genome. Narrow peaks for each strain were identified using MACS2 v2.2.7.1 [168] using the default settings specifically for narrow peak detection. Differential binding was scrutinized first by using BEDTools v2.17.0 [169] closestBed argument and the ASM294v2 annotated feature file.

Once peaks were called, we further scrutinized these peaks to determine fine-scale localization patterns using deepTools [170]. Briefly, Log2 comparisons of each strain to the input were performed using the readCount scale factoring method. Next, our strain comparisons were scaled so that uniform dimensions of 500 bp upstream of the transcription start site, 500 bp downstream of the transcription end site, and region body length of 1500 bp were achieved. After analyzing our peaks from MACS2, we further scrutinized our peaks by performing hierarchical clustering (k = 2) available with deepTools and observed a clear pattern of localization near the promoter region of our genes of interest (Fig 1C). The ChIP-Seq data along with the code for analysis in this study were deposited in the Gene Expression Omnibus with accession number (GSE246596). The wild-type H3K9me2 ChIP-Seq data were published previously and can be accessed publicly in the gene expression omnibus, with accession number GSE119604.

## Co-immunoprecipitation and Western blotting

Co-immunoprecipitation (Co-IP) was carried out as previously described [171]. All Co-IP experiments were performed with a 150 mL starting culture at $\text{OD}_{595} = 1$. Cell pellets were washed with cold 1X PBS and weighed on a fine scale to normalize starting amount. The recommended amount of lysis buffer and glass beads (425–600 μm acid-washed glass beads,

Sigma Aldrich) were added [171]. Pellets were subjected to bead beating 3 times for 45 seconds at 4˚C, and whole cell extracts (WCE) were obtained by tandem spin tube collection. FTP-tagged Lsd1 and Lsd2 were immunoprecipitated using IgG Sepharose beads (GE Healthcare, 17096901) and incubated on an end-over-end rotator for 4–6 hours at 4˚C.

For protein semi-quantitative analysis by western blotting, all cultures were grown to the mid/late exponential growth phase and normalized by spectrophotometry. 4 O.D. of cells were harvested by centrifugation, washed with 1mL of water, and immediately resuspended in 200μL 2X Laemmli Sample Buffer (Bio-Rad, 1610737) with 5% β-mercaptoethanol (BME), and samples were boiled at 95˚C for 5 minutes. Acid-washed glass beads were added, and samples were subjected to 5 minutes of bead beating at 4˚C. Protein extracts were obtained by tandem spin tube collection, and 15μL of each sample was loaded on a denaturing SDS-PAGE gel.

For all Co-IP and western blotting experiments, SDS-PAGE was performed under denaturing conditions using pre-made 4–15% 20μL TGX gels (Bio-Rad, 4561085) for 10 minutes at 100V followed by 35 minutes at 200V. Trans-Blot Turbo system was used to transfer to a 0.2μm PVDF membrane. Membranes were blocked with 4% skimmed milk powder in 1X PBST (1X PBS with 0.1% Tween 20) and washed 3 times for 5 minutes in 1X PBST. Proteins were detected using the appropriate antibody: Peroxidase anti-peroxidase (PAP, 1:1000, Sigma Aldrich) for FTP-tagged proteins, anti-myc antibody (1:1000, Santa Cruz, sc-789), anti-HA antibody (1:1000, Santa Cruz, sc-805), anti-ubiquitin antibody (1:2000, Cell signaling, 3936S), anti-FLAG M2 (1:1000, Sigma Aldrich, F1804) or anti-Mlo3 antibody (1:1000, not commercially available).

## Yeast two-hybrid assay

The yeast two-hybrid assay was performed utilizing the Matchmaker Gold Yeast Two-Hybrid System (TAKARA Bio, USA, 630489), following the standard protocols. The cDNAs of Lsd1, Lsd2, Phf1, and Phf2 were cloned into the pGBKT7 and pGADT7 vectors, respectively. pGBKT7 vectors with fused bait proteins were transformed into Y2H Gold budding yeast cells, and pGADT7 vectors with fused prey proteins were transformed into Y187 budding yeast cells. After mating Y2H Gold and Y187 cells, the diploids were examined on selective plates. pGBKT7-53 and pGADT7-T were used as the positive control, and pGBKT7-Lam was used as the negative control.

## Dilution assay

Indicated strains were resuspended in 1 mL of sterile deionized water and normalized by optical density (O.D.) using a spectrophotometer. 2 O.D. of cell mixture from each strain were transferred to a sterile 96-well plate, followed by ten-fold serial dilutions in sterile water. Cells were spotted using a 48-pin transfer tool onto rich media (YEA), or rich media with 850 μg/mL 5-fluoroorotic acid (5-FOA) for counter selection against uracil expression inserted at the outer terminal repeat region (*otr1R(Sph1)::ura4*+), or other desired medium plates. All plates were incubated at permissive temperature (30˚C) for 2–4 days or until visible colonies appeared in all spots.

## RNA extraction, cDNA synthesis, and qRT-PCR

RNA was extracted from exponentially growing liquid cultures using MasterPure Yeast RNA Purification Kit (Epicentre, MPY03100). During total RNA extraction, samples were incubated at 37˚C for 2–4 hours of DNase I treatment to ensure RNA purity as recommended by the manufacturer. Total RNA was normalized across samples using a micro-volume Nanodrop (Thermo Fisher Scientific, ND-2000), and cDNA synthesis was performed using a M-MLV

reverse transcriptase enzyme kit (Promega, M1701), with the addition of Oligo-dT (Thermo Fisher Scientific, SO131). A no-RT control was additionally performed on each sample to check for genomic DNA contamination by performing a parallel cDNA synthesis on each sample without the addition of the reverse transcriptase. Semi-quantitative RT-PCR was performed by PCR using gene specific primers for 28–30 cycles, and quantitative real-time PCR was performed with SYBR Select Master Mix (Applied Biosystems, 4472908) in triplicates as compared to a constitutively transcribed control gene ($act1^+$ or $leu1^+$ gene, refer to primer list in the S4 Table). At least three replicates of each sample were run, and statistical analysis was performed using a Student's *t*-test. Error bars represent the standard error of the mean.

## Lsd1- and Lsd2-GFP fluorescent imaging

Fresh GFP-tagged *S. pombe* cells growing on SC medium were washed three times in 1XPBS buffer and heat-fixed on glass slides. Images were collected on a Zeiss LSM880 confocal microscope (Carl Zeiss MicroImaging, Thornwood, NY) Airyscan system using a Plan-Apochromat 63×/1.4 oil objective lens. Fixed yeast cells were excited at 488 nm with a 490–640 nm emission range and 2.6% laser power. The frame size was maintained at 1268 x1268 pixels, and the pinhole opening was at 4.77 airy units. The Zeiss ZEN black software was used for image acquisition and Airyscan image processing. Quantification of approximately 100 cells per raw image was conducted using Image J software.

## Ubiquitination assay

The Lsd1-FTP *mts2-1*, Lsd1-FTP *mts2-1 set1Δ*, and Lsd2-FTP *mts2-1* strains were cultured at 26°C overnight and transferred to 33°C for 8 hours. The Lsd1/2-FTP, Lsd1/2-FTP *set1Δ*, and Lsd1/2-FTP *clr4Δ* strains were cultured at 30°C overnight and transferred to 37°C for 2 hours. Cells were harvested from 50 mL cultures at $OD_{595}$ of 0.8, washed with water, and flash-frozen in liquid nitrogen. Thawed cells were resuspended in 150 μL lysis buffer (2 mM $KH_2PO_4$, 8 mM $Na_2HPO_4$, 500 mM NaCl, 2.6 mM KCl, 0.05% NP-40, 10% glycerol, 1 mM DTT, Roche complete protease inhibitor cocktail, and 1 mM PMSF) with 1% SDS [111]. Glass beads were added to grind the cells for homogenization. The cell lysate was obtained by spinning and denatured at 100°C for 10 minutes. After diluting the cell lysate by adding 450 μL lysis buffer, cell debris was removed by centrifugation at 17,530 × g for 10 minutes at 4°C. Pre-cleared cell lysate was incubated with IgG Sepharose beads (GE Healthcare, 17096901) for at least 6 hours. The beads were washed in lysis buffer containing 0.1% SDS at least 4 times and all remaining lysis buffer was removed. Protein samples were eluted by HU loading buffer (8 M urea, 200 mM Tris–HCl pH 6.8, 1 mM EDTA, 5% SDS, 0.1% (wt/vol) bromophenol blue, and 1.5% (wt/vol) DTT) and saved for western blotting [111]. The samples were separated on a 4–12% bis–Tris gel, followed by western blotting using an anti-ubiquitin antibody (Cell Signaling Technology 1:1500, 3936S) to detect ubiquitinated Lsd1 or Lsd2.

## Supporting information

**S1 Fig. Genetic interactions implicate divergent functions of Lsd1 and Lsd2.** (A) Serial dilution assays reveal growth defects in *lsd1-ΔHMG* and *lsd2-ΔC* mutants on rich media (YEA) at the permissive temperature (30°C), along with silencing defects at the outer centromeric repeat (*otr1::ura4⁺*) on 5-FOA. (B-C) Serial dilution assays demonstrate viable genetic interactions between *lsd1-ΔHMG* and *clr4Δ* (B) and between *lsd2-ΔC* and *set1Δ* (C) on rich media (YEA) at the permissive temperature (30°C). (D) A representative tetrad yielded from genetic crosses between *clr4Δ set1Δ* and *lsd1-ΔHMG clr4Δ* (left), or *lsd2-ΔC set1Δ* (right), reveals the lethality of *lsd1-ΔHMG set1Δ clr4Δ* and *lsd2-ΔC set1Δ clr4Δ* triple mutants. (E) qRT-PCR analysis of

peri-centromeric *dg/dh* repeats and *cenH* (mating type locus) demonstrates heterochromatin silencing defects in *lsd1-ΔHMG clr4Δ* and *lsd2-ΔC set1Δ* double mutants, with normalization to wild-type (WT = 1). Asterisks denote significance ($p \leq 0.05$) determined by the Student's *t*-test. Horizontal lines indicate significance between single mutants and double mutants. Error bars represent the standard error of the mean (s.e.m.).
(TIF)

**S2 Fig. Mutant lsd1 protein has slightly higher enrichments at constitutive heterochromatin compared to the wild-type.** (A-B) Loss of silencing at the peri-centromeric region (A) and the mating type locus (B) in *lsd1-ΔHMG* and *lsd2-ΔC* mutants compared to wild-type (WT) using RNA-seq data are illustrated in blue. Overlayed are wild-type Lsd1 or Lsd2 ChIP-Seq data in black and *lsd1-ΔHMG* or *lsd2-ΔC* ChIP-Seq data in red. The chromosome position is shown on the X-axis, and the Y-axis represents the standardized $\text{Log}_2$ ratios for Lsd1/2 RNA-Seq and ChIP-Seq (range from -2 to 2) and H3K9me2 ChIP-Seq (range from 0 to 30). H3K9me2 ChIP-Seq showing the heterochromatic regions are highlighted in green. Zoomed-in regions of the red box on the left panel are displayed on the right (A-B). Figures were generated using the Integrated Genome Browser (IGB).
(TIF)

**S3 Fig. Lsd1 and Lsd2 robustly bind to the *sah1*+ promoter.** ChIP-Seq peaks showcase the localization of Lsd1/2 at the *sah1*+ promoter region, with *sah1*+ schematic denoted on the X-axis. Arrows besides the *sah1*+ *gene* pinpoint the oligo positions mentioned in Fig 1H. The Y-axis displays standardized (range from 0 to 6) $\text{Log}_2$ fold enrichments to untagged control. This figure was generated using the Integrated Genome Browser (IGB).
(TIF)

**S4 Fig. Lsd2-BD and Phf1-BD strains exhibit self-activation of reporter genes without requiring a Gal4 activating domain (AD).** (A-B) When combined with an empty vector as a negative control in the yeast two-hybrid system, Lsd1-AD, Lsd1-BD, Lsd2-AD, Phf2-BD, Phf1-AD, and Phf2-AD do not induce reporter gene expression. (C-D) Lsd2-BD and Phf1-BD autonomously triggered the expression of reporter genes when combined with an empty AD vector. These colonies exhibited growth on two selective plates: (A & C) medium lacking leucine and tryptophan but containing X-α-gal (-Leu -Trp +X-α-gal) and (B & D) medium without adenine, leucine, and tryptophan (-Ade -Leu -Trp).
(TIF)

**S5 Fig. Lsd1 directly interacts with Phf1 and Phf2, but Lsd2 lacks direct interactions with Lsd1 or Phf2.** (A-B) Serial dilution assays show direct interactions between Lsd1 with Phf1 and Phf2. In contrast, Lsd2 does not engage in direct interactions with either Lsd1 or Phf2. Colonies on two selective plates confirm these interactions: (A) medium lacking leucine and tryptophan but supplemented with X-α-gal (-Leu -Trp +X-α-gal) and (B) medium without adenine, leucine, and tryptophan (-Ade -Leu -Trp).
(TIF)

**S6 Fig. Clr4 and Set1 deletions affect Lsd1 enrichments at different genomic loci.** ChIP-Seq peaks for Lsd1 at specified genetic backgrounds reveal its localization in two distinct loci on each chromosome. The X-axis shows the genomic position on each indicated chromosome. The Y-axis represents standardized $\text{Log}_2$ fold enrichments (range from 0 to 8) compared to the untagged control. This figure was made using the Integrated Genome Browser (IGB).
(TIF)

**S7 Fig. Deletions of Clr4 and Set1 have distinct effects on Lsd2 enrichments at various genomic sites.** The ChIP-Seq peaks for Lsd2 in indicated genetic contexts highlight its positioning at two specific loci on each chromosome. The X-axis indicates the genomic position on each indicated chromosome, and the Y-axis represents standardized $Log_2$ fold enrichments (range from 0 to 8) relative to the untagged control. This figure was generated in the Integrated Genome Browser (IGB).
(TIF)

**S8 Fig. The removal of Clr4 and Set1 has opposing effects on the levels of Lsd1- and Lsd2-GFP.** (A-B) Lsd1-GFP (A) and Lsd2-GFP (B) in indicated genetic backgrounds were imaged using a Zeiss LSM880 confocal microscope. (C-D) Quantification of approximately 100 cells per raw image was conducted using Image J software. Lsd1-GFP (C); Lsd2-GFP (D). Asterisks denote significance ($p \leq 0.05$) determined by the Student's *t*-test. Horizontal lines indicate significance between wild-type and mutants. Error bars represent the standard error of the mean (s.e.m.).
(TIF)

**S9 Fig. The variations in Lsd protein levels cannot be attributed to changes in mRNA levels.** (A -B) RNA-Seq analysis compares normalized RNA-Seq reads (RPKB) aligned with the genomic locus of Lsd1 (A) and Lsd2 (B) in *clr4Δ* or *set1Δ* backgrounds to those in the wild type. The data reveals that the loss of Clr4 or Set1 has a minimal impact on the overall mRNA levels of *lsd1+* and *lsd2+*. The graphical representation was generated using the Integrated Genome Browser (IGB). (C-D) Quantitative RT-PCR analysis of *lsd1+* (C) and *lsd2+* (D) mRNA levels reinforce the observation that changes in mRNA levels in *clr4Δ* and *set1Δ* do not align with variations in Lsd1 and Lsd2 protein levels when normalized to wild-type (WT = 1). Asterisks indicate statistical significance ($p \leq 0.05$) as determined by the Student's *t*-test when comparing the indicated samples with WT values. Error bars represent the standard error of the mean (s.e.m.).
(TIF)

**S10 Fig. During heat stress, Lsd1 and Lsd2 transcripts show minimal changes.** (A-B) We analyzed normalized RNA-Seq reads (RPKB) aligned to the genomic loci of *lsd1+* (A) and *lsd2+* (B) in wild-type cells at both 30˚C and 37˚C. Additionally, we assessed the mRNA levels of *lsd1+* and *lsd2+* in a *set1Δ* background at 37˚C. Graphs were made using the Integrated Genome Browser (IGB). (C-D) qRT-PCR analysis of *lsd1+* mRNA levels (C) and *lsd2+* mRNA levels (D) demonstrate the impact of heat stress on Lsd1/2 transcription, with or without Set1, normalized to wild-type (WT = 1). Statistical significance ($p \leq 0.05$) is indicated by asterisks, determined by the Student's *t*-test when comparing the indicated samples with WT values. Error bars represent the standard error of the mean (s.e.m.).
(TIF)

**S1 Table. RNA-seq data for *set1Δ*, *lsd1-ΔHMG* and *lsd2-ΔC* under 37˚C.**
(XLSX)

**S2 Table. Gene Ontology of most differently expressed genes under 37˚C.**
(XLSX)

**S3 Table. List of strains used in this study.**
(PDF)

**S4 Table. List of oligonucleotides used in this study.**
(PDF)

**S1 Data. Supplementary data related to Figs 1C, 2G and 2H (combined peaks).**
(XLSX)

**S2 Data. Supplementary data related to Figs 1C, 2G and 2H (combined clusters).**
(XLSX)

**S3 Data. Supplementary data related to Fig 1H.**
(XLS)

**S4 Data. Supplementary data related to Fig 4H.**
(CSV)

**S5 Data. Supplementary data related to S8 Fig.**
(XLSX)

**S6 Data. Supplementary data related to S9 Fig.**
(XLSX)

**S7 Data. Supplementary data related to S10 Fig.**
(XLSX)

## Acknowledgments

We thank Glen Marrs and Umma Fatema for helping with the Confocal Microscope. We appreciate comments from Gloria Muday, Sarah Esstman, Regina Cordy, and David Ornelles.

## Author Contributions

**Conceptualization:** Haoran Liu, Bahjat Fadi Marayati, Ke Zhang Reid.

**Data curation:** Haoran Liu, David de la Cerda, Brendan Matthew Lemezis, Ke Zhang Reid.

**Formal analysis:** Haoran Liu, Bahjat Fadi Marayati, David de la Cerda, Brendan Matthew Lemezis, Ke Zhang Reid.

**Funding acquisition:** Ke Zhang Reid.

**Investigation:** Haoran Liu, Bahjat Fadi Marayati, Brendan Matthew Lemezis, Jieyu Gao, Qianqian Song, Minghan Chen, Ke Zhang Reid.

**Methodology:** Haoran Liu, David de la Cerda, Brendan Matthew Lemezis, Jieyu Gao, Ke Zhang Reid.

**Project administration:** Ke Zhang Reid.

**Resources:** Ke Zhang Reid.

**Software:** David de la Cerda, Qianqian Song, Minghan Chen.

**Supervision:** Ke Zhang Reid.

**Validation:** Haoran Liu, Bahjat Fadi Marayati, Jieyu Gao, Ke Zhang Reid.

**Visualization:** Ke Zhang Reid.

**Writing – original draft:** Haoran Liu.

**Writing – review & editing:** Haoran Liu, David de la Cerda, Brendan Matthew Lemezis, Jieyu Gao, Ke Zhang Reid.

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
