## [Decision Letter · Decision Letter 0]

15 Sep 2023

Dear Dr Reid,

Thank you very much for submitting your Research Article entitled 'The Cross-Regulation Between Set1, Clr4, and Lsd1/2 in Schizosaccharomyces pombe' to PLOS Genetics.

The manuscript was fully evaluated at the editorial level and by independent peer reviewers. The reviewers appreciated the attention to an important problem, but raised some substantial concerns about the current manuscript. Based on the reviews, we will not be able to accept this version of the manuscript, but we would be willing to review a much-revised version. We cannot, of course, promise publication at that time.

If you decide to revise the manuscript for further consideration at PLOS Genetics, please aim to resubmit within the next 60 days, unless it will take extra time to address the concerns of the reviewers, in which case we would appreciate an expected resubmission date by email to plosgenetics@plos.org.

We are sorry that we cannot be more positive about your manuscript at this stage. Please do not hesitate to contact us if you have any concerns or questions.

Yours sincerely,

Songtao Jia

Guest Editor

PLOS Genetics

Wendy Bickmore

Section Editor

PLOS Genetics

Reviewer's Responses to Questions

**Comments to the Authors:**

Reviewer #1: Liu et al. demonstrated the cross-regulation between histone-modifying enzymes. To my best knowledge, this cross-regulation has never been described before, and this finding will help understand how distinct posttranslational modifications of histones intimately play a role in shaping the transcriptome. However, at least to me, it is not clear enough how biologically significant the cross-regulation is. Also, some experiments need to be done to conclude.

(1) Based on the genetic interactions shown in Fig. 2A, I’d say that Lsd1 and Lsd2 primarily target H3K9me and H3K4me, respectively. If this description is correct, both clr4Δset1Δlsd2-ΔC and clr4Δset1Δlsd1-ΔHMG should be viable. I wonder if this speculation is correct. If clr4Δ or set1Δ fail to suppress the inviability of Δset1Δlsd2-ΔC or clr4Δlsd2-ΔC, respectively, I'd like to know the plausible reason why the triple mutants are not viable. I believe the additional genetic interaction analyses will further strengthen the conclusion.

(2) WB results such as Fig. 3A and 3B, there are more than one band of Lsd1 and Lsd2. These multiple bands are specific to tagged proteins, as untagged strain samples did not show any signal. What is the nature of those bands? Any posttranslational modifications?

(3) The changes in the steady-state levels of Lsd1 and Lsd2 in mutants (e.g., clr4Δ and set1Δ) are clear, but have you tested transcription rate, mRNA levels, translation efficiency (or ribosome occupancy on mRNA), or protein stability? I suspect the changes result from the increased or decreased protein half-life, and I did not see such analyses (or did I miss them?!)

(4) I see the cross-regulation between the three protein complexes does exist. However, I am afraid I still do not quite understand the significance of the cross-regulation. What kind of advantage would the cross-regulation have? For example, can bulk histone methylation assessed by WB be maintained even in lsd mutants?

(5) As to the Y2H assays, was there any particular reason why you did not spot serial dilutions of budding yeast? The data shown in S3 Fig are qualitative, but data with the serial dilutions are more quantitative. Or is qualitative data sufficient for your conclusion?

(6) line 101 and S Tables, "the 5’ end" and "5’ FW" should be "the 5´ end" and "5´ FW" Use "prime" not "apostrophe.”

Reviewer #2: In this manuscript, Liu and colleagues investigate the crosstalk regulation between Set1 (H3K4methyltransferase), Clr4 (H3K9methyltransferase) and the lysine demethylases Lsd1 and Lsd2 in the model fission yeast. By performing genetic, biochemical and omics approaches the authors show that C-terminal truncations of Lsd1 and Lsd2 do not affect their complex composition but impair their chromatin binding, while also demonstrate that Clr4 and Set1 have opposite effects on Lsd1 and Lsd2 protein levels but not on transcripts. Importantly, such regulation is achieved through the ubiquitin-proteasome-dependent pathway. In addition, the authors show that Lsd1/2 protein levels increase upon heat stress and that Clr4 associated complex (CLRC) regulates Set1 levels. Overall, the data presented is convincing and increase our understanding of how lysine demethylases can be regulated. Please find below my comments which would improve this already fine manuscript.

_ While centromeric and mating-type silencing are readily detectable in the lsd1 and lsd2 mutant cells, it is not clear whether the mutant Lsd1 and Lsd2 proteins still localize to such regions. The authors shall consider including these data. I also wonder whether the upregulation at cenH can be noticeable in cells harboring Kint2:ura4+ marker or, alternatively, mat2P upregulation (i.e. haploid meiosis) in REIIΔ cells.

_ Discussion: Despite the fact that PLoS Genetics does not have length restriction, discussion is well written but it seems to be very long for the standard reader and therefore it should be shorten.

_ By performing Western blots, the authors clearly show that Clr4 and Set1 modulate the protein levels of Lsd1 and Lsd2 in opposite ways. It would be nice to demonstrate such regulation by employing a microscopical approach (e.g. GFP-tagging Lsd1 and Lsd2). This is not compulsory, but it would improve the ms.

_ Y2H experiments: The authors state “Lsd2 does not physically interact with Lsd1, Phf1, or Phf2”. I guess the authors mean direct interaction as Co-IP clearly shows such interactions. In addition, for the technical reasons explained in Fig S3 there is no available data for Lsd2-Phf1 interaction.

_ Fig. 2G-H and lines 295-298: The commented differences for Lsd1 and Lsd2 enrichments are not robust. Can the authors show a couple of specific examples that illustrate this point?

Minor comments

Fig. 1H: It is not clear what antibody was used in ChIP.

S2 Fig: Nomenclature (i.e. Lsd1+, Lsd2+)

Line 142: Typo “In S. Pombe”

Line 243: Define the product of Ade2 (i.e. Phosphoribosylaminoimidazole carboxylase, involved in adenine biosynthesis)

Lines 263-265: Based on the data shown, it looks like set1Δ is epistatic to lsd2-ΔC. So, I am not sure if this can be defined as a positive interaction.

Line 401: Typo “ubiqitin”

Reviewer #3: Histone-modifying enzymes play critical roles in gene expression and chromatin organization. In fission yeast, Set1 is the sole H3K4 methyltransferase while Clr4 is the H3K9 methyltransferase. In addition, fission yeast has also two highly conserved H3K4 and H3K9 demethylases, Lsd1 and Lsd2. In this manuscript, the authors found that Lsd1/2 interacts with the Clr4 complex (CLRC) and Set1-associated complexes (COMPASS). The further showed that Clr4 and Set1 modulate the protein levels of Lsd1 and Lsd2 in opposite ways through the ubiquitin-dependent pathway. In addition, the authors showed that Set1 mediates Lsd1/2 level during heat stress. The results of this work are interesting. The experiments in the manuscript are generally well designed and executed. However, there are some concerns, especially regarding evidence to support the conclusion, as listed below:

Major

1. To convincingly show that Clr4 and Set1 mediate Lsd1 and Lsd2 through the ubiquitin-dependent pathway, the in vitro ubiquitination assays are needed.

2. Assays in Fig 4S should be repeated to show any significant difference between WT and the mutants. Alternatively, RT-qPCR can be used.

3. Line 365-367, Figure 2C and 2D: without Set1, Lsd1/2 protein levels are lowly expressed during heat stress. But at 30°C, Lsd1 in the set1 mutant is already very low. The authors may want to check if Lsd1 transcription is affected in the mutant by RT-qPCR.

4. Line 374-375: “we only observed a slight decrease in ubiquitination of Lsd1and Lsd2 in the clr4Δ background”. I don’t find a slight decrease in ubiquitination of Lsd2. In my view, it is even a little bit of an increase from the figure.

5. Line 238-247: the Y2H figures need more controls. Firstly, the cells grown on SD-LT plates indicating both plasmids have been transformed to the cells is necessary. In addition, the negative control of each BD or AD-containing protein with another empty plasmid are necessary in the same figure to show there is no self-activation in the Y2H assay.

6. In some WB figures, like fig. 2c, fig.4 there are multiple bands. Please label the right one in the figure, so the readers can know them easily.

Minor:

Line 142: change “S. Pombe” to “S. pombe”.

Line 174: please indicate which region is the amine oxidase domain located?

Line 177: Please briefly explain the function of HMG domain.

Fig. 2I & 3: Please indicate which protein do you use as control in the fig? In addition, it indicates the Lsd1 and Lsd2 bands in the Fig. 2I.

Line 383: change “wildtype” to “wild-type”.

**Have all data underlying the figures and results presented in the manuscript been provided?**

Reviewer #1: Yes

Reviewer #2: Yes

Reviewer #3: None

PLOS authors have the option to publish the peer review history of their article (what does this mean?). If published, this will include your full peer review and any attached files.

Reviewer #1: No

Reviewer #2: No

Reviewer #3: No

---

## [Decision Letter · Decision Letter 1]

27 Nov 2023

Dear Dr Reid,

Thank you very much for submitting your Research Article entitled 'The Cross-Regulation Between Set1, Clr4, and Lsd1/2 in Schizosaccharomyces pombe' to PLOS Genetics.

The manuscript was fully evaluated at the editorial level and by independent peer reviewers. The reviewers appreciated the attention to an important topic but identified some concerns that we ask you address in a revised manuscript.

We therefore ask you to modify the manuscript according to the review recommendations. Your revisions should address the specific points made by each reviewer.

Yours sincerely,

Songtao Jia

Guest Editor

PLOS Genetics

Wendy Bickmore

Section Editor

PLOS Genetics

The revised manuscript was evaluated by reviewer 1 and reviewer 3. While they are mostly satisfied with the responses to their comments raised during the first round of review, reviewer 3 still has some concerns about the in vitro ubiqutination assays. Given that you do not plan to include the data in the manuscript, it is prudent to acknowledge the possibility that the effect of CLRC on Lsd1/2 proteins may be indirect, as suggested by reviewer 3. While you have acknowledged the possible indirect effect in your concluding sentence, you can also add more  discussion of possible indirect effects in Page 26, in the section "The role of CLRC and COMPASS in Lsd1/2 segregation via UPS".

Reviewer's Responses to Questions

**Comments to the Authors:**

Reviewer #1: The authors satisfactorily addressed the comments I raised. This manuscript by Liu et al. should be accepted, and I expect to read a follow-up work on non-histone substrates.

Reviewer #3: The authors addressed most of concerns. The only remaining major concern is the in vitro ubiquitination assays. The quality of the in vitro ubiquitination data that the authors provided is poor; in addition, the figure legend is confusing: does figure (A) show just a western blot or ubiquitination assays? Furthermore, I cannot find the figure legend for figure (C). Unless the authors can provide more convincing data, at minimum, they should acknowledge the possibility that the effect of CLRC on Lsd1/2 proteins may be indirect in the manuscript.

Minor point:

About the Y2H in Figure 1J and 1K, the new version includes proper controls, which clarified my concerns. The minor point is that they didn't mention what plates they used; they just labeled +x-a-gal and -Adenine in the figure. Please provide more detailed information about the medium (is it -his, -Leu, -Trp?)

**Have all data underlying the figures and results presented in the manuscript been provided?**

Reviewer #1: Yes

Reviewer #3: None

PLOS authors have the option to publish the peer review history of their article (what does this mean?). If published, this will include your full peer review and any attached files.

Reviewer #1: No

Reviewer #3: No

---

## [Editor Report · Decision Letter 2]

12 Dec 2023

Dear Dr Reid,

We are pleased to inform you that your manuscript entitled "The Cross-Regulation Between Set1, Clr4, and Lsd1/2 in Schizosaccharomyces pombe" has been editorially accepted for publication in PLOS Genetics. Congratulations!

Yours sincerely,

Songtao Jia

Guest Editor

PLOS Genetics

Wendy Bickmore

Section Editor

PLOS Genetics

Comments from the reviewers (if applicable):

**Data Deposition**

http://datadryad.org/submit?journalID=pgenetics&manu=PGENETICS-D-23-00858R2

**Press Queries**

---

## [Editor Report · Acceptance letter]

29 Dec 2023

PGENETICS-D-23-00858R2 

The Cross-Regulation Between Set1, Clr4, and Lsd1/2 in Schizosaccharomyces pombe 

Dear Dr Reid, 

We are pleased to inform you that your manuscript entitled "The Cross-Regulation Between Set1, Clr4, and Lsd1/2 in Schizosaccharomyces pombe" has been formally accepted for publication in PLOS Genetics! Your manuscript is now with our production department and you will be notified of the publication date in due course.

With kind regards,

Lilla Horvath

PLOS Genetics

On behalf of:
